# Endocytosis-mediated entry of a caterpillar effector into plants is countered by Jasmonate

Zi-Wei Yan[1,2,7], Fang-Yan Chen[1,2,3,7], Xian Zhang[1,2], Wen-Juan Cai[4], Chun-Yu Chen[1,2], Jie Liu[1,5], Man-Ni Wu[1,2], Ning-Jing Liu[3], Bin Ma[3], Mu-Yang Wang[1], Dai-Yin Chao [ID][3], Cai-Ji Gao [ID][6] & Ying-Bo Mao [ID][1] ✉

Insects and pathogens release effectors into plant cells to weaken the host defense or immune response. While the imports of some bacterial and fungal effectors into plants have been previously characterized, the mechanisms of how caterpillar effectors enter plant cells remain a mystery. Using live cell imaging and real-time protein tracking, we show that HARP1, an effector from the oral secretions of cotton bollworm (*Helicoverpa armigera*), enters plant cells via protein-mediated endocytosis. The entry of HARP1 into a plant cell depends on its interaction with vesicle trafficking components including CTL1, PATL2, and TET8. The plant defense hormone jasmonate (JA) restricts HARP1 import by inhibiting endocytosis and HARP1 loading into endosomes. Combined with the previous report that HARP1 inhibits JA signaling output in host plants, it unveils that the effector and JA establish a defense and counterdefense loop reflecting the robust arms race between plants and insects.

To defend against potential threats, plants use specialized receptors that detect molecular patterns associated with pathogens, herbivores, and damage (PAMPs/HAMPs/DAMPs) to trigger an array of defense responses[1–7]. Jasmonate (JA), a lipid-based plant hormone, acts as a key regulator of these defenses against insects and pathogens[8–11]. Fatty Acid-Amino Acid Conjugates (FACs) are widely known elicitors in insect oral secretion (OS) that are reported to significantly amplify the JA-burst in plants undergoing insect herbivory[1,2]. Besides, many other herbivore associated molecular patterns (HAMPs) are able to active plant defense response[12] such as inceptins, the proteolytic fragments of chloroplastic ATP synthase γ-subunit in the OS of *Spodoptera frugiperda* larvae, promote the JA and JA-Ile accumulation in cowpea plants[7,13]. To suppress their host defenses, insects and pathogens secrete small protein molecules known as effectors which can interfere with many aspects of the immune system[12,14]. An early reported insect

effector, Glucose Oxidase (GOX), was identified from the corn earworm (*Helicoverpa Zea*) salivary gland and inhibits the wounding-induced nicotine accumulation in tobacco[15,16]. Most of the reported insect effectors are from piercing-sucking insects[17,18]. C002-like proteins were first discovered in *Acyrthosiphon pisum* saliva and exist in various aphid species which influence the aphid feeding behaviors and contribute to successful infestation[19,20]. Bt56 is an effector found in whitefly OS which promotes insect infection in tobacco[21]. In addition to the usual protein molecules, RNA can also act as an effector. For example, the noncoding RNA *Ya* of *Myzus persicae* can be delivered into plants through an aphids stylet during feeding and can migrate systemically to suppress plant immunity[22].

The entry of heterogenous regulators into host plants are usually achieved by diffusion, passing through transporters in the plasma membrane, or by secretion systems like exocytosis and endocytosis

[1]CAS Key Laboratory of Insect Developmental and Evolutionary Biology, CAS Center for Excellence in Molecular Plant Sciences (CEMPS), Institute of Plant Physiology and Ecology (SIPPE), Chinese Academy of Sciences (CAS), Shanghai, China. [2]University of CAS, Shanghai, China. [3]National Key Laboratory of Plant Molecular Genetics, CEMPS/SIPPE, CAS, Shanghai, China. [4]Core Facility Center of CEMPS/SIPPE, CAS, Shanghai, China. [5]Shanghai Key Laboratory of Plant Molecular Sciences, College of Life Sciences, Shanghai Normal University, Shanghai 200234, China. [6]Guangdong Provincial Key Laboratory of Biotechnology for Plant Development, School of Life Sciences, South China Normal University (SCNU), Guangzhou, China. [7]These authors contributed equally: Zi-Wei Yan, Fang-Yan Chen. ✉e-mail: ybmao@cemps.ac.cn

trafficking[23–26]. Many bacterial pathogens used specialized secretion machinery to directly inject effectors into host cells[27]. The RxLR-like effectors of oomycetes bind to the phosphatidylinositol-3-phosphate (PI3P) phospholipid and enter host cells through lipid raft-mediated endocytosis[28]. Effectors can also be transported via extracellular vesicles (EVs), which transfer substances such as proteins, RNAs, lipids, and metabolites[29–31]. Patellins (PATLs) and Tetraspanins (TETs) are types of membrane proteins that are also found in exosomes, suggesting that they are involved in cell-to-cell trafficking[32]. *Botrytis cinerea* infection induces plant secretion of TET8-related exosomes, which carry plant-derived sRNAs and trigger cross-kingdom RNAi[33]. Choline transporter-like 1 (CTL1) is another component in vesicle trafficking system, which is involved in auxin efflux transporters also known as PIN-FORMED proteins (PINs) distribution[34].

When aphids feed on or probe a plant, they deliver effectors into host cells directly through their stylets. Once inside the plant, the effectors can travel to nearby cells or become localized in specific intracellular targets. For example, some effectors from the green peach aphid (*Myzus persicae*) have different final locations, with Mp10 located in the cytoplasm and chloroplast of mesophyll cells while MpPIntO1 and MpC002 reside in the sheath-like structures created by the insects' stylet near the feeding site[35]. This specificity of localization indicates that the translocation of insect effectors in plants is highly regulated, however, the mechanisms of their import into plants are largely unknown. In our previous work, we identified an effector known as HARP1 from cotton bollworm (*Helicoverpa armigera*) OS which suppresses JA response in plants[36]. HARP1-like proteins are widely present in Lepidoptera and their functions are likely conserved in the family Noctuidae. In this study, we demonstrate that HARP1 enters plant cells through endocytosis and that the vesicle trafficking-related proteins CTL1, PATL2, and TET8 are required for its entry. Interestingly, we found that endocytosis-mediated entry of HARP1 into plants is countered by JA, which restricts the import by inhibiting endocytosis and HARP1 loading into endosomes, thereby creating a counter-defense response.

## Results

### HARP1 is granulated and is moving in plants

In our previous work, we found that HARP1 from the cotton bollworm interacts with JAZs which are the repressors in JA signaling to interfere with plant wounding responses[36]. To trace HARP1 transportation in plants, we generated fluorescent protein fused HARP1 (Venus-HARP1, V-HARP1) for visualization. Using confocal microscopy, we observed V-HARP1 around the leaf wounding sites of *Arabidopsis* while Venus alone was barely detected. We also found that this transportation is dose associated, with higher concentrations of V-HARP1 leading to stronger fluorescence intensity in the leaf cells. At the moderate concentration (0.1 mg/ml), the V-HARP1 signals could be still detected in cells after incubation, although the fluorescence appeared weaker than in that of the 1 mg/ml concentration (Supplementary Fig. 1a). For optimal observation, we choose a concentration of 1 mg/ml, which is also the concentration commonly used to study fungal effector transportation[28]. Our results show that application of V-HARP1 to a leaf wounding site impairs the inductions of gene expressions involved in the plant wounding response (Supplementary Fig. 1b). This indicates that the effector activity is not affected by the N-terminal fusion of Venus. In addition to *Arabidopsis*, we successfully introduced V-HARP1 into a variety of plants including cotton, tobacco, and rice callus (Supplementary Fig. 2). In *Arabidopsis* leaves, V-HARP1 appears as granules with an average size of 1.2 μm² (Fig. 1b and Supplementary Fig. 4a). We observed that about 44% of the total V-HARP1 granules were moving (Supplementary Fig. 4b and Supplementary Movie 1). This hints that HARP1 cellular entry may be mediated by vesicle trafficking. The entry of HARP1 into plant cells was further confirmed by transmission electron microscopy (TEM) observation. V-HARP1 was found in the pavement and mesophyll tissues of *Arabidopsis* while Venus alone could not be observed (Fig. 1a and Supplementary Fig. 3).

### HARP1 localizes in endosomes

To exclude the possibility that the punctate appearance of V-HARP1 in plant cells was caused by the fused Venus, we observed how Venus is distributed when alone. Since Venus is unable to get into plant cell like V-HARP1, we independently injected both Venus and V-HARP1 into leaf tissue. We found that most of the Venus proteins were uniformly distributed while V-HARP1 was observed as granules (Supplementary Fig. 5). Furthermore, we indicated internalized endosomes using an endocytosis tracer called N-(3-Triethylammoniumpropyl)-4-(6-(4-(diethylamino)phenyl)hexatrienyl) pyridinium dibromide (FM4-64)[37,38]. Our results show that some V-HARP1 granules colocalized with FM4-64-stained endosomes (Supplementary Fig. 5). We observed a similar colocalization when V-HARP1 was incubated in wounded leaves (Supplementary Fig. 6). A portion of these V-HARP1-loaded endosomes were also observed to be moving (Supplementary Movies 2, 3).

The early endosome (EE) is the first station of the endocytosis mediated vesicle trafficking which is undistinguishable with the trans-Golgi network (TGN)[39,40]. mCherry-fused clathrin light chain 2 (CLC2) and mRFP-fused vacuolar proton ATPase subunit VHA-a isoform 1 (VHA-a1) are used to mark the clathrin-coated vesicles (CCVs) and secretory vesicles (SVs) of the TGN/EE, respectively[39,41,42]. In addition to colocalizing with both CCVs and SVs, V-HARP1 was found to be further transported into the prevacuolar compartment/multivesicular body/late endosome (PVC/MVB/LE) as illustrated by its colocalization with mCherry fused PVC/MVB/LE marker Rab homolog F2A (Rha1)[43–45] (Fig. 1b, c and Supplementary Fig. 7). Roughly 50%, 85%, and 35% of V-HARP1 molecules colocalized with CCVs, SVs, and PVC/MVB/LEs, respectively (Fig. 1d). These results indicate that HARP1 located in TGN and was further sorted.

### HARP1 import is mediated by endocytosis

The endosome recycling inhibitor, brefeldin A (BFA) blocks vesicle trafficking from the TGN to PM, leading to the enlarged endosomes called BFA bodies[46] due to the TGN aggregations[47]. When BFA-pretreated *Arabidopsis* leaves were incubated with V-HARP1, fluorescent signals were observed around the wounding sites, mirroring the results of the mock samples (Fig. 2a, b). However, that V-HARP1-loaded endosomes tended to aggregate after BFA treatment (Fig. 2c). Tyrphostin A23 (A23) and Wortmannin (Wm)[47,48] are endocytosis inhibitors. We found the pretreatment of these inhibitors largely reduced the fluorescent signals of V-HARP1 around the wounding sites (Fig. 2a, b). Furthermore, though V-HARP1 was able to merge with the FM4-64 labeled PM, there was a substantial decline in the punctate patterns of V-HARP1, and granulated V-HARP1 was hardly detected by A23 or Wm pretreatments (Fig. 2c, d). Taken together, these results suggest that cellular V-HARP1 import requires endocytosis.

In the immunoblot assay, the imports of both V-HARP1 and the native HARP1 were reduced to an undetectable level by the pretreatment of A23 (Fig. 2e). Furthermore, the whole amount immunohistochemistry reveals that the signals of HARP1 from OS-incubated leaves were much weaker in the A23 pretreated samples compared to those A23 free samples (Fig. 2f). These results suggest that the import of native HARP1 depends on endocytosis, which is consistent with the import of recombinant V-HARP1.

### Cellular import of V-HARP1 is reduced in some vesicle trafficking mutants

CTL1, Sphingolipids Delta-8 desaturase (SLD), PATLs, and TETs regulate endomembrane systems and are involved in maintaining membrane lipid homeostasis and vesicle trafficking[32–34,48–54]. In this study,

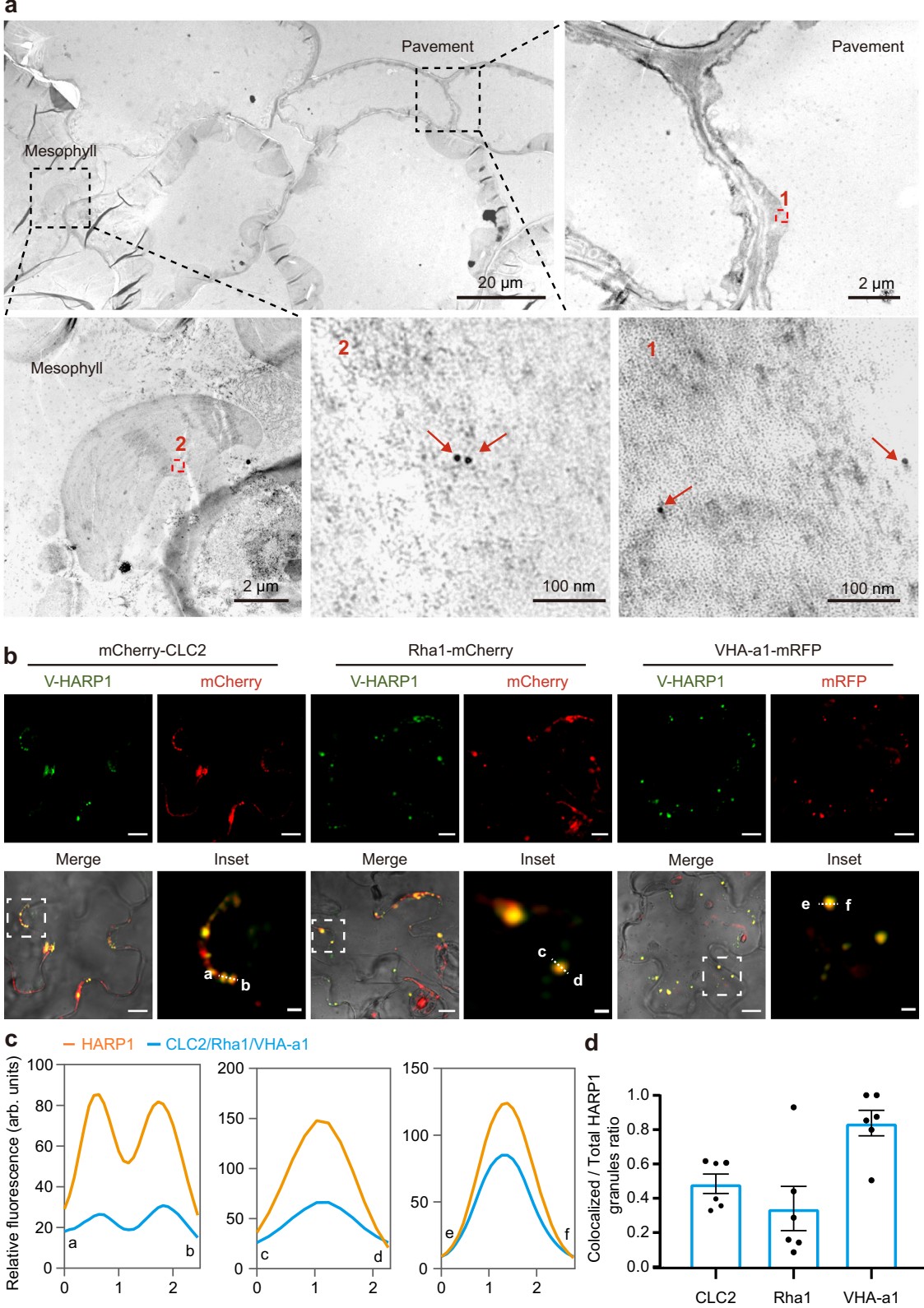

we analyzed whether HARP1 import is affected in mutants with altered vesicle trafficking components. We found that V-HARP1 import was significantly reduced in *ctl1*, *sld1 sld2*, *patl2*, and *tet8*, but was barely affected in *35 S::PDLP5*, *patl1*, and *patl3* (Fig. 3a, b and Supplementary Fig. 8). Considering that PDLP5 (Plasmodesmata-located protein 5) negatively regulates the permeability of plasmodesmata by enhancing callose deposition and overexpressing PDLP5 (*35 S::PDLP5*)[55], these

results exclude the possibility that V-HARP1 is predominantly imported via the plasmodesmata. When purified recombinant V-HARP1 was injected into leaves of the wild type (Col-0), *ctl1*, *patl2*, and *tet8*, it appeared as granules. However, these granules were largely reduced in *ctl1*, *patl2*, and *tet8* (Fig. 3c, d). This indicates that vesicle trafficking-related proteins CTL1, PATL2, and TET8 are responsible for HARP1 granulation in plant leaves.

**Fig. 1 | V-HARP1 gets into pavement and mesophyll cells and locates in endosomes.** The wounded plant leaves of wild-type (**a**), or of mCherry-CLC2, Rha1-mCherry and VHA-a1-mRFP transgenic *Arabidopsis* (**b**) were incubated with V-HARP1, respectively. **a** Transmission electron microscope (TEM) observation of V-HARP1 in pavement and mesophyll cells. Ultrathin sections from V-HARP1 treated leaves were immune-gold labeled with anti-GFP antibody. The enlarged view of the red boxed regions (1 and 2) from the observed pavement and mesophyll cells was shown independently. The arrows indicate the immune-gold labeled V-HARP1. Scale bars were indicated in each image. **b** V-HARP1 colocalized with the mCherry-CLC2 labeled clathrin-coated vesicles, the VHA-a1-mRFP labeled secretory vesicles and the Rha1-mCherry labeled prevacuolar compartment/multivesicular body or late endosomes. The inset panels highlighted the dotted square regions. Scale bar: 2 μm for inset and 10 μm for others. **c** Fluorescence intensity (in arbitrary units, arb. units) in cross-section [dotted line in inset (**b**)]. The orange indicate the intensities of V-HARP1, and the bluelines indicate the intensities of CLC2, Rha1 and VHA-a1. **d** Quantification of CLC2, Rha1 or VHA-a1 colocalized V-HARP1. Ratio of the colocalized to total V-HARP1 granules in (**b**) were calculated. Data are mean ± SEM (*n* = 6), 6 leaves from 3–5 independent plants were examined. Source data are provided as a Source Data file.

HARP1 must be successfully imported into a plant cell before it can interact with JAZ proteins and suppress the induction of JA response genes. We tested the V-HARP1 effects on the wounding response in *ctl1*, *patl2*, and *tet8*. In general, wounding could induce the expressions of JA response genes, but the inductions were somewhat reduced. When V-HAPR1 was applied to the leaf wounding sites, the inductions of defense gene expressions were significantly reduced in the wild type but were hardly affected in *ctl1*, *patl2*, and *tet8* (Fig. 3e–g). Moreover, the expressions of *CTL1*, *PATL2*, and *SLD2* were induced by V-HARP1 application to the leaf wounding sites when compared with Venus application (Supplementary Fig. 9). Feeding test showed enhanced insect resistance in *ctl1* and *patl2* but not in *tet8*. Interestingly, *35S:PDLP5* even displayed reduced resistance (Supplementary Fig. 10). This is possibly explained by these genes having impacts on multiple physiological processes in plants.

## HARP1 targets CTL1, PATL2, and TET8 for successful import

CTL1 and TET8 are transmembrane proteins and PATL2 is a membrane-associated protein. TET8 and PATL2 are also both found in exosomes[31–33]. So, these proteins have the possibility to directly bind with HARP1 in extracellular face. We then examined this inference. In the Bimolecular Luciferase Complementation (BiLC) assay, we determined that full-lengths of CTL1, PATL2, and TET8 can interact with HARP1 while PATL1 and PATL3 cannot (Supplementary Fig. 12b, c). As transmembrane proteins, the outside membrane domains of CTL1 and TET8 are likely able to bind to HARP1. The predicted structure of CTL1 contains five extracellular loops (ECs), of which EC1 is the longest with 182 amino acids[56] (Supplementary Fig. 11a). TET8 contains a very short EC1 loop and a longer EC2 loop with 139 amino acids[31] (Supplementary Fig. 11b). PATL2 has a C-terminal GOLD domain (PATL2C110) which is predicted to mediate protein-protein interactions[50]. Our Yeast two-hybrid (Y2H) assays revealed that HARP1 can interact with CTL1EC1, PATL2C110, and TET8EC2 (Supplementary Fig. 12a). Consistently, our pull-down assay illustrated that CTL1EC1, PATL2C110, and TET8EC2 can be co-immunoprecipitated with HARP1 (Fig. 4a).

Similar to V-HAPR1, HARP1 fused at the N-terminal with dsRed (dsRed-HARP1) and mCherry (mCherry-HARP1) fluorescent proteins is also able to enter leaf cells through wounding sites (Supplementary Fig. 13). To confirm the interaction between HARP1 and CTL1, dsRed-HARP1 was incubated with wounded *pCTL1::CTL1-GFP* leaves. We found that roughly 65% of dsRed-HARP1 colocalized with CTL1-GFP inside plant cells (Fig. 4c–e), further providing in vivo evidence for the HARP1-CTL1 interaction.

To know which domain of HARP1 is required for interaction with CTL1, PATL2, and TET8, we tested the binding activity of truncated HARP1 (Supplementary Fig. 14a) with CTL1EC1, PATL2C110, and TET8EC2. A Y2H assay revealed that the deletion of five amino acids in the HARP1 C-terminal (V-HARP1δC5) and of 44 or more in the N-terminal (V-HARP1δN44 and V-HARP1δN49) abolished the its interaction with CTL1EC1, PATL2C110, and TET8EC2. However, the deletion of 39 or fewer amino acids in the HARP1 N-terminal (V-HARP1δN34 and V-HARP1δN39) had no obvious effects on these protein interactions (Supplementary Fig. 14b–d). Consistent with these results, our pull-down assay demonstrated that V-HARP1δN39 can be co-immunoprecipitated with CTL1EC1, PATL2C110, and TET8EC2, while V-HARP1δN44 cannot (Fig. 4b). We then determined the imports of truncated HARP1 into plant cells and found that V-HARP1δN34 and V-HARP1δN39 but not V-HARP1δC5, V-HARP1δN44, and V-HARP1δN49 can be successfully imported (Fig. 4f and Supplementary Fig. 15). Interestingly, when V-HARP1δN39 and V-HARP1δN44 were injected into plant leaves, only the former was observed as granules (Supplementary Fig. 14e, f). These results suggest that both amino acid residues 40-49 and 117-122 are required for the HARP1 interaction with CTL1, PATL2, and TET8. Next, we tested the effector activity of truncated V-HARP1 and showed that the induction of wounding response genes is suppressed by V-HARP1δN34 and V-HARP1δN39 but not by V-HARP1δN44 treatments (Fig. 4g). This indicates that the interactions of HARP1 with CTL1, PATL2, and TET8 proteins are required for the effectors successful import into host plant cells.

## Plant defense hormone JA counters HARP1 import

It is well known that in animals, an immune response is required to inhibit an invading foreign protein[57]. In a similar fashion, HARP1 import into plants may be restricted by host immunity. As the main defense hormone, JA and JA-Ile become highly elevated during plant responses to insect herbivory and mechanical wounding[58–60]. The JA synthesis deficient mutant, *aos*, displays a largely reduced wounding response[61,62]. Analysis of the transcriptome data of the wild type and *aos* detected around 92 genes with annotated responses to JA (Supplementary Data file 2). Consistently, the expressions of these genes in the wild type were more highly elevated 2 h post-wounding than those in *aos*. 37 of these JA response genes displayed significantly higher expressions in wild type than in *aos*, while only three were the opposite (Supplementary Fig. 16a, b). However, in response to wounding, other genes (1121) exhibited higher expressions in *aos* than in the wild type (Supplementary Fig. 16c) and were significantly enriched in GO items related to the secretory vesicle, membrane, cell wall, extracellular region, and endoplasmic reticulum (Fig. 5a). The ClueGO network showed that the genes related to the cell wall, extracellular region, and endoplasmic reticulum were closely associated with membrane related items (Fig. 5b). Consistent with the RNA-seq analysis, qRT-PCR revealed the expressions of four selected genes (EXLA2, GRP-5, EARLI1, and PKS2) encoding membrane related proteins which were more highly induced in *aos* plants than in the wild type (Fig. 5c). Together, these results suggest that JA may negatively associate with membrane related functions. Since membrane proteins are key players in endocytosis and vesicle homeostasis[63–65], we analyzed whether JA impacts endocytosis. Normally, the internalized endosomes traced by FM4-64 were less prevalent in the wounded leaves of the wild type than those of *aos*. When in the presence of MeJA, endosome accumulation is reduced to a similar degree between wild type and *aos* (Fig. 5d). These results indicate that JA negatively regulates endocytosis.

Next, we investigated whether HARP1 import is affected by JA. Following incubation in wounded leaves, the fluorescence signal of V-HARP1 was found to be weaker in the JA hypersensitive mutant *jazQ*, stronger in *aos*, and similar in the JA insensitive mutant *coi1-2* when compared with that in the wild type (Fig. 6a and Supplementary

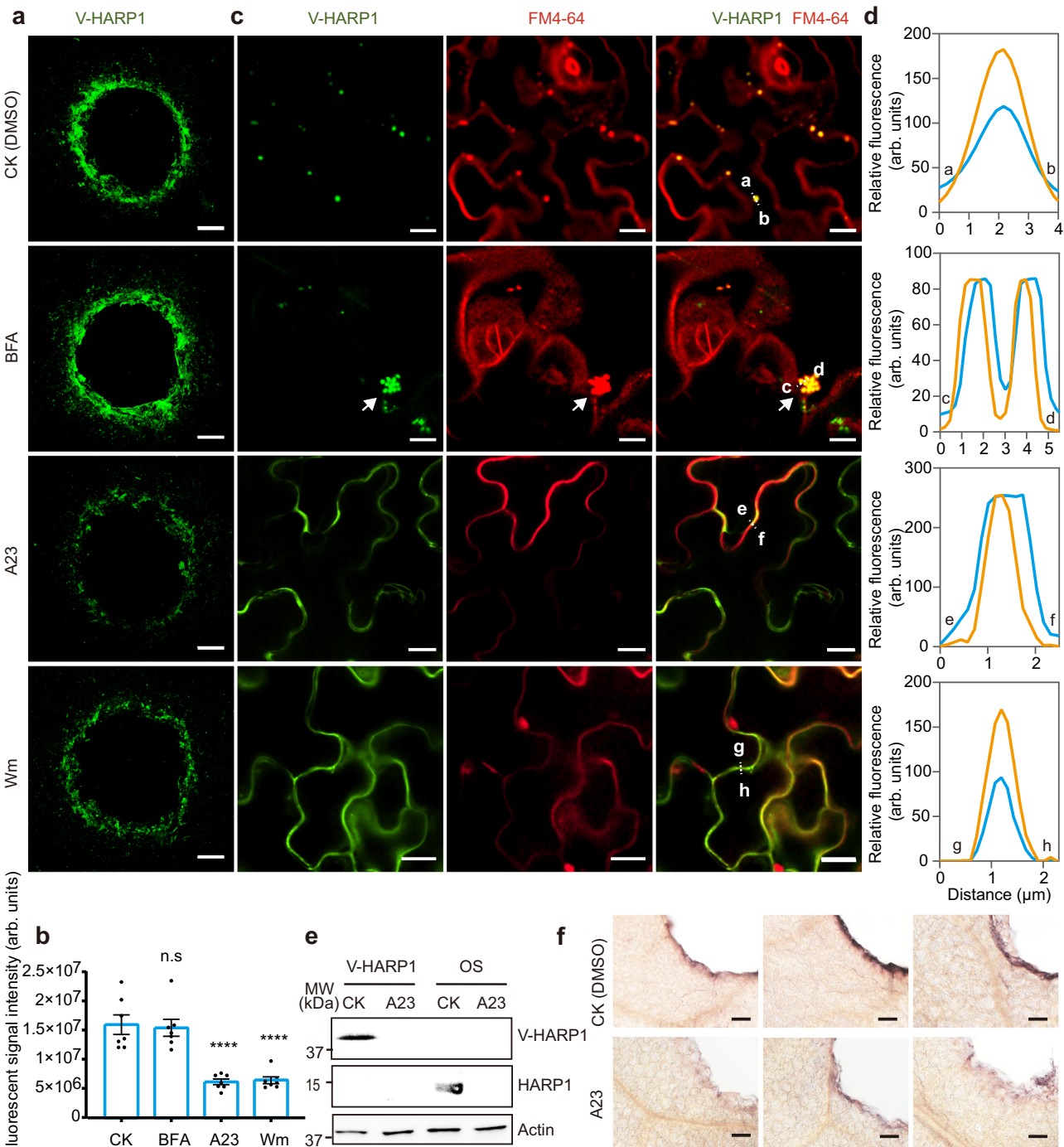

**Fig. 2 | The entry of HARP1 into plant cells is mediated by endocytosis.** The wounded *Arabidopsis* leaves were pretreated with DMSO (CK), A23, Wm and BFA before incubated with V-HARP1. **a** V-HARP1 signals were reduced in the leaves with A23 and Wm pretreatment. Scale bar: 200 μm. **b** Quantification of the fluorescent signal intensity (in arbitrary units, arb. units) of V-HARP1 in (**a**). Data are mean ± SEM (*n* = 7). Seven leaves from 4–5 independent plants were examined. Data are analyzed by one-way ANOVA with two-sided Dunnett's post hoc test (n.s: not significant, ****$p < 0.0001$). **c** BFA pretreatment caused the aggregation of V-HARP1-located endosomes and the granulated V-HARP1 inside the pavement cells of the A23 and Wm pretreated leaves were disappeared. The arrow indicates the aggregation of V-HARP1-located endosomes. FM4-64 was used to trace internalized endosomes. Scale bar: 10 μm. **d** Fluorescence intensity (in arbitrary units, arb. units) in cross-section [dotted line in (**c**)]. The orange lines indicate the intensity of V-HARP1, the blue lines indicate the intensity of FM4-64 in leaves pretreated with DMSO (CK), BFA, A23 and Wm, respectively. **e** Detection of HARP1 in leaf discs by immunoblot assay. The wounded leaf discs were pretreated with DMSO (CK) and A23 before incubated independently with V-HARP1 solutions and the OS of cotton bollworm larvae. Anti-HARP1 antibody was used to detect V-HARP1 and HARP1. **f** Whole amount immunohistochemistry detection of HARP1 at the wounding sites. The OS-incubated leaf discs as described in (**e**) were used for assay. Anti-HARP1 antibody was used to detect HARP1. Scale bar: 100 μm. Source data are provided as a Source Data file.

Fig. 17). In the MeJA pretreatment, the V-HARP1 signal was significantly reduced in both the wild type and *aos*, but was not notably affected in *coi1-2*. In *jazQ*, the V-HARP1 signals were low with or without the MeJA pretreatment (Fig. 6a and Supplementary Fig. 17).

When focusing on endosomes in the wounding treatment, we found that the counts of both total and V-HARP1-loaded endosomes were higher in *aos* and *coi1*-2 when compared to the wild type. During the combination of wounding and MeJA treatment, the amount of total

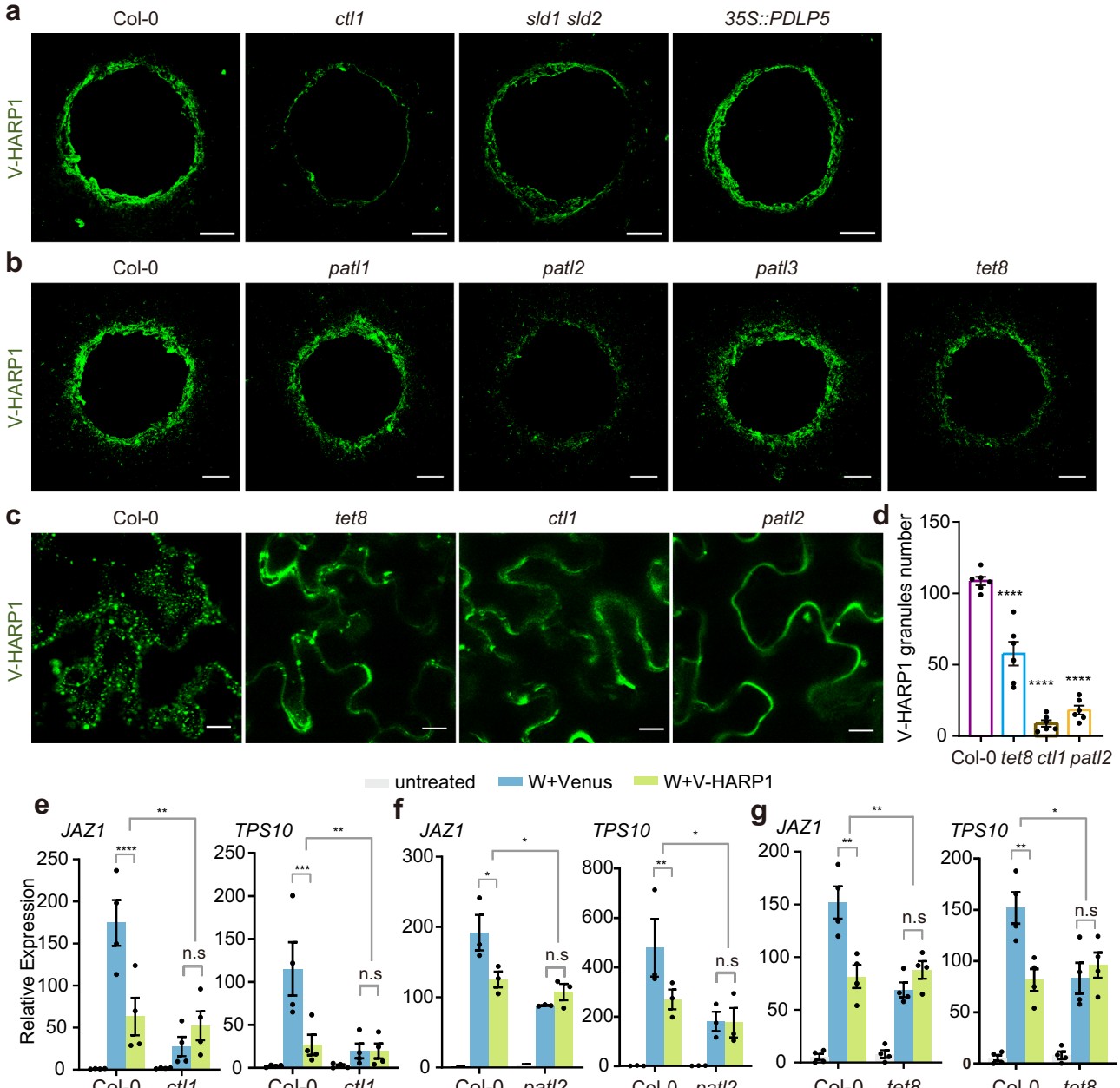

**Fig. 3 | Deficient V-HARP1 import is companied with the reduced effector activities. a, b** V-HARP1 was less imported into *ctl1*, *sld1 sld2*, *patl2* and *tet8*. Leaves of the wild type (Col-0) and indicated mutants were wounded and incubated with V-HARP1. Scale bar, 200 μm. **c** Injected V-HARP1 in plants were observed by confocal microscopy. V-HARP1 (0.2 μg/ml) was injected into wild type (Col-0), *tet8*, *ctl1* and *patl2*. Scale bar, 5 μm. **d** Quantification of the V-HARP1 granules in (**c**). The leaves were selected to count the number of V-HARP1 granules. Data are mean ± SEM (*n* = 6 biological replicates) and analyzed by one-way ANOVA with two-sided Dunnett's post hoc test (****p < 0.0001). **e–g** V-HARP1 reduced wounding response in wild type (Col-0) but not in *ctl1* (**e**), *patl2* (**f**) and *tet8* (**g**). Plant leaves were wounded and quickly applied with 1 mg/ml Venus (W+Venus, blue indicated) or V-HARP1 (W + V-HARP1, green indicated) proteins on the wounding sightes, respectively. Gene expressions 4 h post treatments were detected by qRT-PCR. The gene expressions in the unwounded plants (untreated, gray indicated) were set to 1. Data were analyzed by two-way ANOVA followed by multiple comparisons with two-sided Fisher's LSD test (n.s: not significant, *P < 0.05, **P < 0.01, ***P < 0.001, ****P < 0.0001). Data are mean ± SEM (*n* = 4 biological replicates in **e**, **g** *n* = 3 biological replicates in **f**). Source data are provided as a Source Data file.

and V-HARP1-loaded endosomes was dramatically reduced in the wild type and *aos* plant, while no significant change was observed in *coi1-2*. In *jazQ*, endosome counts were extremely low in both consequences (Fig. 6b-d). Interestingly, the proportion of V-HARP1-loaded to total endosomes significantly dropped in wild type and *aos* after MeJA pretreatment (Fig. 6e).

JAR1 catalyzes the conjugation of JA with isoleucine to produce JA-Ile, the active molecule in JA signaling. In *jar1*, the deficiency of generating JA-Ile caused an insensitive response to JA treatment. As

shown in supplementary fig. 18, both the HARP1 entry and endosome accumulations in *jar1* were not strongly affected by the presence of MeJA. Taken together, these results suggest that JA signaling negatively regulates cellular HARP1 import by inhibiting endocytosis and HARP1 loading into endosomes.

## Discussion

Although plants have sophisticated physical barriers, insects can force their effectors into plant cells. While many studies have focused

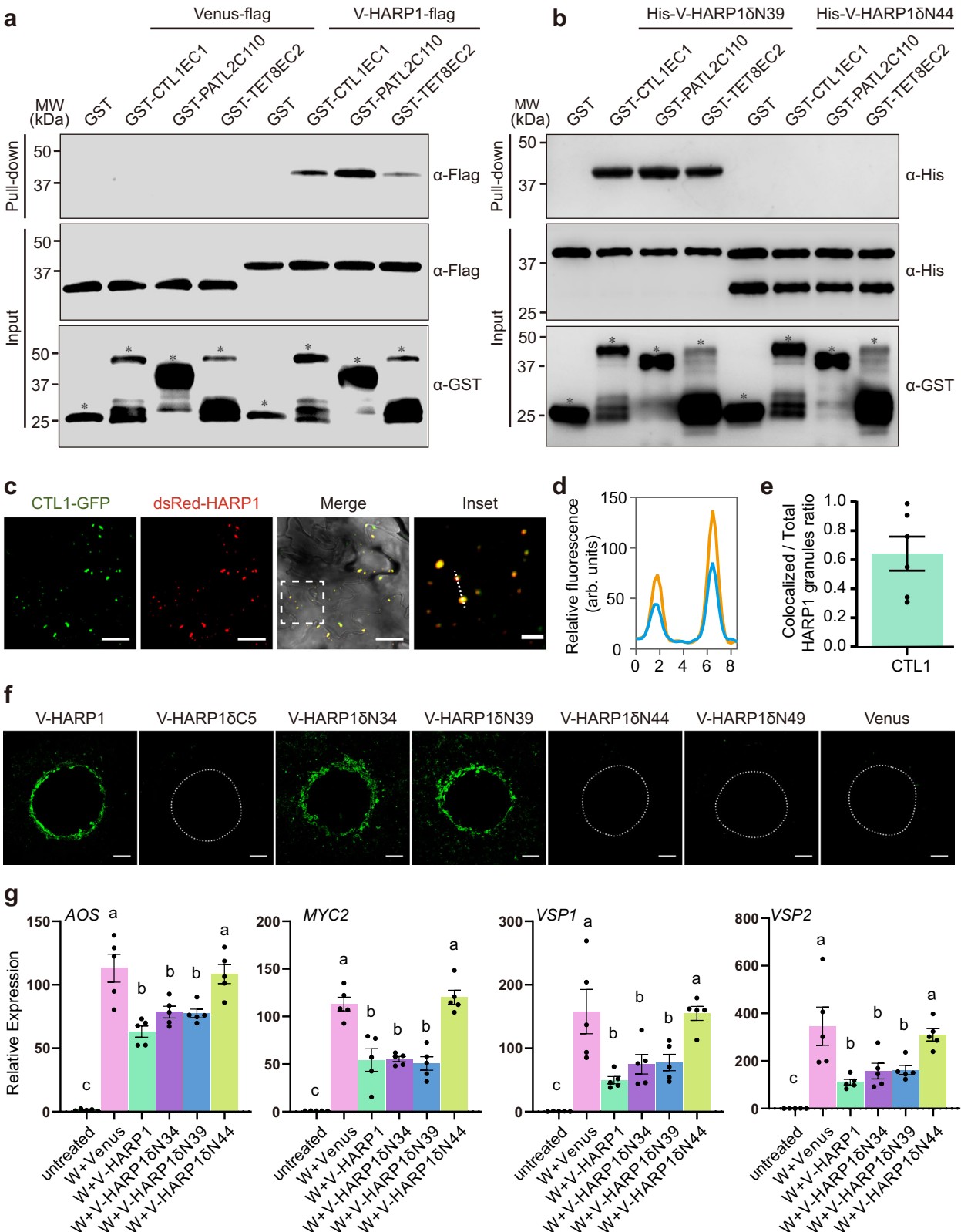

on the mechanisms of effector-inhibited plant immunity or defense responses, very little work has been done to understand the import process of effectors into plant cells. Here, we found that the *H. armigera* effector HARP1 enters plant cells via endocytosis. We also determined that its capacity for trafficking depends on its binding activities with CTL1, PATL2, and TET8. These interactions likely also internalize HAPR1 into plant cells. PATL2 and TET8 have been found

in extracellular vesicles (EV) and are involved in cell-to-cell trafficking[32,33,51], indicating the possibility of HARP1 intercellular transport. After entering plant cells, HARP1 further interacts with JAZs to block JA defense responses against insect herbivory. On the other hand, JA established a counter-defense response by inhibiting endocytosis and HARP1 loading into endosomes to restrict import (Fig. 7). Although some JAZs can interact with HARP1, JAZ was mainly

**Fig. 4 | The interactions of HARP1 with CTL1, PATL2 and TET8 are required for its successful import. a, b** Pull-down assay of the full length (**a**) and truncated (**b**) V-HARP1 interacting with GST fused CTL1EC1, PATL2C110 and TET8EC2. Recombinant GST fused proteins were incubated with the total leaf proteins of *N. benthamiana* transiently expressing Venus-flag or V-HARP1-flag, respectively (**a**) or with recombinant proteins of His-V-HARP1δN39 and His-V-HARP1δN44 (**b**). Anti-GST antibody was to detect GST fused proteins in input. Anti-flag antibody was used to detect flag fused Venus and V-HARP1 (**a**). Anti-His antibody was used to detect the His fused truncated V-HARP1 (**b**). Free GST and GST fused proteins were asterisked. **c** dsRed-HARP1 colocalized with CTL1-GFP. The wounded *Arabidopsis* leaves of *pCTL1::CTL1-GFP* were incubated with dsRed-HARP1 and the pavement cells were observed under confocal microscopy. The inset panel highlights the dotted square region of the merged channel. Scale bar: 5 μm for inset and 20 μm for others. **d** Fluorescence intensity (in arbitrary units, arb. units) of the cross sections [dotted line in inset (**c**)]. Blue and orange lines indicate CTL1-GFP and dsRed-HARP1

fluorescence signals, respectively. **e** Quantification of the CTL1-GFP colocalized HARP1 granules in (**c**). The proportions of the colocalized granules in total were calculated. 6 leaves from 3-5 independent plants were examined. Data are mean ± SEM (*n* = 6). **f** Imports of truncated V-HARP1 into leaf tissues. The wounded *Arabidopsis* leaves were incubated with purified V-HARP1 and its truncated forms (Supplementary Fig. 15). The dotted-lines indicated the wound sites. Scale bar, 200 μm. **g** Effects of V-HARP1 and its truncated variants on *Arabidopsis* wounding response. Leaves were wounded and then painted with full length or truncated V-HARP1 on the wounding sites quickly. The expressions of the indicated genes were detected 4 h post treatments by qRT-PCR. The gene expressions in the unwounded plants (untreated) were set to 1. Data are mean ± SEM (*n* = 5 biological replicates) and analyzed by one-way ANOVA followed by two-sided Tukey's post hoc test for multiple comparisons. Different letters indicate significant differences (*P* < 0.05). Source data are provided as a Source Data file.

located in nucleus, this reduced the possibility to meet HARP1 outside of cells. And for the JAZs of damaged cells which may be presented in extracellular spaces, wounding will quickly trigger their degradation. Based on these facts, JAZ is unlikely to mediate HARP1 import directly. These findings reveal the role of JA in the arms race between plants and insects. In animals, the invading proteins are monitored and removed by the immune system. In plants, the entry of insect effectors is restricted by JA signaling, reflecting similarities between plant and animal immune responses.

In plants, vesicle trafficking is involved in the transport of toxic substances to resist pests and pathogens[66,67]. Here, we found that the insect hijacked the plant vesicle trafficking system to distribute their own effectors. This supports the idea that both insects and microbial pathogens use similar strategies to deliver effectors into host plant cells. The RxLR-like effectors of filamentous pathogens like oomycetes and fungi enter plant cells by lipid raft-mediated endocytosis[28]. Conserved motifs like RXLR, YKARK, and RGD are required for their successful entry[68]. It is speculated that the cellular import of this type of effector could be mediated by certain membrane-associated proteins or transporters, but this had yet to be identified[69]. Unlike these pathogen effectors, we found that no single motif is responsible for HARP1 transportation. Nevertheless, both residues 40-49 and 117-122 are required for successful HARP1 import (Fig. 4f), highlighting the differences between the transportation of insect and pathogenic microbe effectors in plants.

The introduction of exogenous molecules into plant cells is often required in plant research and utilizes methods like *Agrobacterium* transformations and gene guns. These molecules can also be introduced with the help of cell-penetrating peptides and nanoparticles[70,71]. A particularly unique study used the fungal pathogen *Ustilago maydis* to deliver a bioactive host protein into the plant to alter maize anther cell behavior in Situ[72]. In our study, we discovered that HARP1 fused with Venus can be delivered into the cells of various plant species, suggesting that it is possible to develop a HARP1-based delivery system for exogenous molecules import into plants.

## Methods
### Plant materials and growth conditions
All the *Arabidopsis* used in this research were under the background of ecotype Col-0. The PIP2A-RFP[73], VHA-a1-mRFP[39], mCherry-CLC2[42], Rha1-mCherry[34], *35 S::PDLP5*[49], *pCTL1::CTL1-GFP*[48], *coi1-2*[74], *jazQ*[75], *ctl*[48], *sld1 sld2*[49], *tet8*[54], *aos*[61], *patl2*[76], *patl3*[51] were described previously. And *patl1* (SalK_056190.2) were obtained from the *Arabidopsis* Biological Resource Center (ABRC).

The seeds of *Arabidopsis thaliana*, tobacco plants (*Nicotiana benthamiana*) and cotton plants (*Gossypium hirsutum*) used in this study were grow in long days (16-h light/8-h dark) under 22℃.

The rice (*Oryza sativa*) seeds were washed with 75% ethanol for 1 min, and with 20% (v/v) NaClO for 30 min, then rinsed five times with

sterile water. The mature embryos were grown on $N_6D_2$ medium and cultured in dark at 28 °C for 14 days to induce callus.

### Plant treatment
HARP1 (XP_047035071) was identified from the oral secretions of cotton bollworm (*Helicoverpa armigera*)[36] and the sequence information can be found in NCBI (https://www.ncbi.nlm.nih.gov/protein/XP_047035071.1?report=genbank&log$=protalign&blast_rank=1&RID=F3YF3FK8016). To detect the effector activity of V-HARP1, the second pair of true leaves at the rapid expanding stage (about 18-day-old seedlings) were wounded (punched 3 - 4 holes per leaf) and the unwounded leaves were used as negative control, the V-HARP1 and Venus protein solutions (1 mg/ml) were painted to the wounding sites immediately. Four hours later, leaves were harvested and used for qRT-PCR analyses of the indicated wounding-induced gene expressions.

To observe the import of HARP1 in plants, the leaves of *Arabidopsis*, cotton and tobacco were cut into small pieces with a hole in the center and incubated with V-HARP1 or its truncated forms (1 mg/ml) for 2–4 h. Samples were then washed with PBS containing 0.08% BSA for at least 5 times, 20 min for each time and detected under confocal microscopy. As a negtive control, the Venus (1 mg/ml) were used for incubation with plant leaves like V-HARP1 and followed by the same washing procedure to insure get rid of all the extra proteins adhering to the surface and subsequently for confocal microscopy. Under our experimental condition, Venus along cannot be detected in leaf cells.

For endocytosis inhibition assay, endocytosis inhibitors of Tyrphostin A23 (A23, Sigma-Aldrich), Wortmannin (Wm, Sigma-Aldrich) and Brefeldin A (BFA, TCI) were dissolved in Dimethyl sulfoxide (DMSO, Sigma-Aldrich). The wounded *Arabidopsis* leaves were incubated with 50 mM Tris-HCl (pH 8.0) containing 30 μM A23, 30 μM 30 Wm and 50 μM BFA, respectively and leaves incubated with 50 mM Tris-HCl (pH 8.0) containing equal amount of DMSO were used as mock treatment. After half an hour, V-HARP1 was added and incubated for another 2–4 h before confocal microscopy assay.

For the assay of jasmonate effects on V-HARP import, MeJA (Sigma-Aldrich) was dissolved in ethanol and diluted in double-distilled water to a final concentration of 50 μM. As the mock treatment, water solutions with equal volumes of ethanol were used. Eighteen-day-old *Arabidopsis* were sprayed with water solutions of MeJA and ethanol, respectively. Two hours later, detached leaves were punctured and incubated with the V-HARP1 solutions for 2–4 h and then observed under confocal microscopy.

### Confocal microscopy
Venus was fused to the N-terminal of HARP1 (V-HARP1) for visualization. After incubation with Venus or V-HARP1 followed by 4–5 times washing, samples were detected by confocal microscopy. To detect HARP1 in cells, regions 100 - 600 μm from the wounding site were selected for observation (Supplementary Fig. 19). We found that the

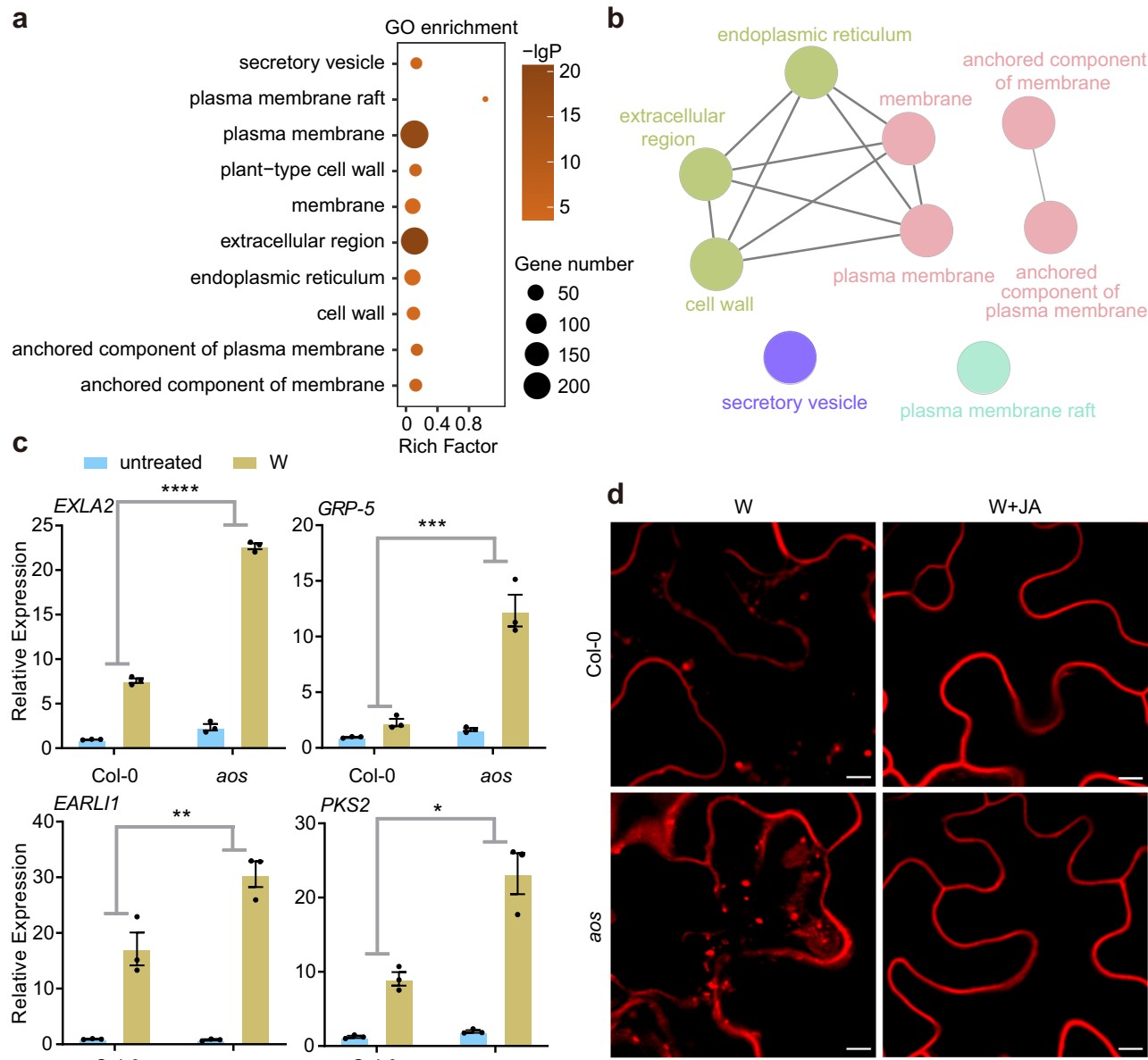

**Fig. 5 | JA negatively regulates plant endocytosis. a–c** JA inhibits the expressions of genes related to secretary vesicles and membrane. Ten-day-old *Arabidopsis* (Col-0) and *aos* were wounded and leaves were harvested 2 h post wounding (W). The untreated plant leaves were used as control. Total RNAs were extracted for RNA sequencing and qRT-PCR analysis. **a** GO enrichment of the genes with expression significantly higher in *aos* than those in wild type (Col-0) after wounding (*aos*_W > Col-0_W). **b** The ClueGO network of the enriched items as described in (**a**). **c** qRT-PCR of the selected genes (*aos*_W > Col-0_W) according to the RNA-seq data. The gene expressions in the unwounded wild type (Col-0) (untreated) were set to 1.

Data are mean ± SEM (*n* = 3 biological replicates) and analyzed by two-way ANOVA followed by multiple comparisons with two-sided Fisher's LSD test. (\**p* < 0.05, \*\**p* < 0.01, \*\*\**P* < 0.001, \*\*\*\**P* < 0.0001). Blue and yellow box indicate untreated and wounding (W) treatment, respectively. **d** JA suppressed the accumulations of internalized endosomes. The wounded leaves of Col-0 and *aos* were incubated with 50 μM MeJA (W + JA) or Ethanol (W) solutions and further stained with FM4-64 to trace the internalized endosomes. Pavement cells were observed under microscopy. Scale bar, 5 μm. Source data are provided as a Source Data file.

fluorescent signal from the Venus treated samples were hardly to detect, therefore, excluding auto fluorescing which might be caused by wounding and the possibility that Venus itself could enter plant cells under our experimental condition. To label plasma membrane and trace the internalized endosomes, samples were incubated with 2 μM FM4-64 (Invitrogen) in the dark for 5–7 min, and washed twice with double-distilled water.

All images and movies were taken by confocal systems, Leica SP8 or Olympus spinSR equipped with different immersive objectives (Leica SP8 with 20XW/NA 0.75 and 60XW/NA 1.2, Olympus spinSR with 30XSil/NA 1.05 and 60XSil/NA 1.30). The confocal z-axis resolution range is about 1240 - 647 nm. The moving HARP1 granules were traced

by time series at a fixed layer. For multi-channel images, we use sequential imaging procedure to exclude fluorescence signals crosstalk.

The excitation/emission wave lengths for YFP and FM4-64/mRFP/mCherry signals were 515/525-560 and 561/610-650 nm, respectively. And Images were captured with strictly identical acquisition parameters for the quantitative fluorescence intensity. The fluorescence signal intensity was calculated using ImageJ software and statistically analyzed with GraphPad Prism software.

**Immuno-localization**

After the *Arabidopsis* leaves were incubated with Venus or V-HARP1 followed by 4–5 times washing, samples were cut into pieces (1–2 mm²)

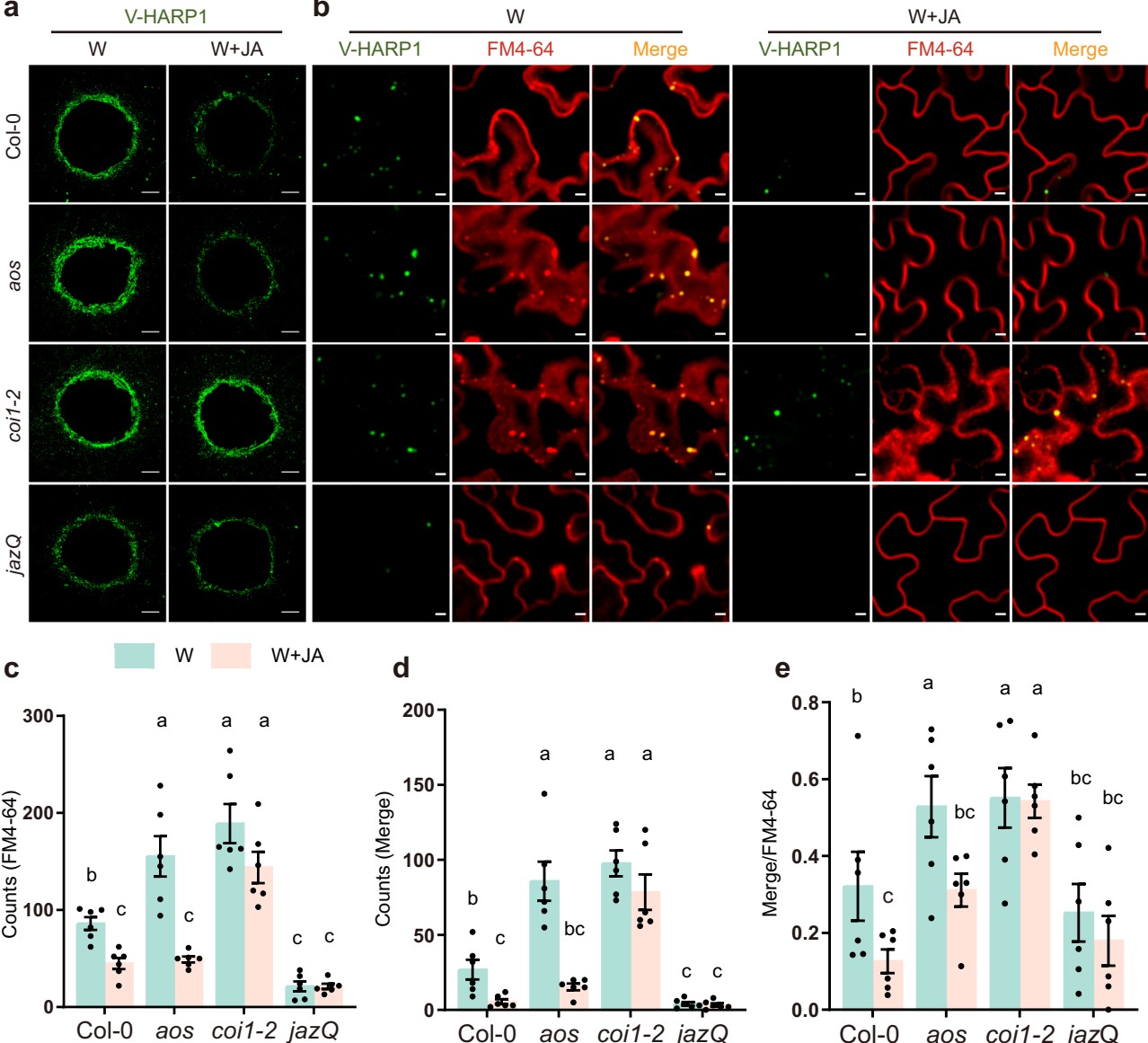

**Fig. 6 | JA counters HARP1 import into *Arabidopsis*. a** V-HARP1 signals around the wounding sites of plant leaves. The indicated wounded leaves of *Arabidopsis* were pretreated with Ethanol (W) or 50 µM MeJA (W + JA) for 2 h and subsequently incubated with V-HARP1. Scale bar: 200 µm. **b** V-HARP1-loaded endosomes in the plants. Plants were treated as described in (**a**). The internalized endosomes were traced by FM4-64. Scale bar: 5 µm. **c**–**e** Quantification of endosomes from (**b**). The leaves were selected to count the number of endosomes. Light blue and light pink box indicate Ethanol (W) or 50 µM MeJA (W + JA) treatment, respectively. Data are mean ± SEM ($n = 6$ biological replicates) and analyzed by two-way ANOVA followed by multiple comparisons with two-sided Fisher's LSD test. Different letters indicate significant differences ($P < 0.05$). FM4-64 stands for FM4-64 traced endosomes (**c**); merge stands for V-HARP1-loaded (**d**) endosomes and merge/FM4-64 stands for the ratio of V-HARP1-loaded to total endosomes (**e**). Source data are provided as a Source Data file.

and immediately fixed in 4% (w/v) paraformaldehyde in phosphate-buffered saline (PBS) at 4 °C overnight. Fixed leaves were embedded in LR White resin (Sigma-Aldrich) after dehydration through a graded alcohol series. Sections were prepared on a Leica Microsystem UC7 ultramicrotome. Ultrathin sections of ~70 nm were mounted onto nickel grids for immuno-gold labeling.

Immuno-gold labeling was performed as follows: 3% acetylated bovine serum albumin C in PBST (PBS + 0.05% Tween 20) (BSA/PBST) for 1 h; anti-GFP antibody (1:50 dilution in 3% BSA/PBST) for 4 h; PBST sixteen times for 2 min; goat anti-mouse IgG conjugated with 10-nm colloidal gold particles (1:50 diluted in 3% BSA/PBST, Sigma-Aldrich) for 2 h; PBST ten times for 2 min; and water twice for 2 min. Grids were observed under 120 kV transmission electron microscope (TEM) after staining. The replicate sections of Venus

treated samples had no signals of immuno-gold in cells indicating that the signal is specific and Venus alone could not enter plant cells.

### Insect feeding test

The cotton bollworm (*Helicoverpa armigera*) larvae were obtained from the Institute of Zoology, Chinese Academy of Science. The newly hatched larvae were raised on artificial diet[77] till they were used for assay. About 12–16 seedlings of *Arabidopsis* were planted in one pot. Five to six pots of three-week-old plants of wild type (Clo-0), *tet8*, *ctl1*, and *patl2* were used for insect feeding test. About 5–6 third instar larvae were put into one pot and covered with a plastic cover. After feeding for 4 days, larvae were weighted individually (Supplementary Fig. 20).

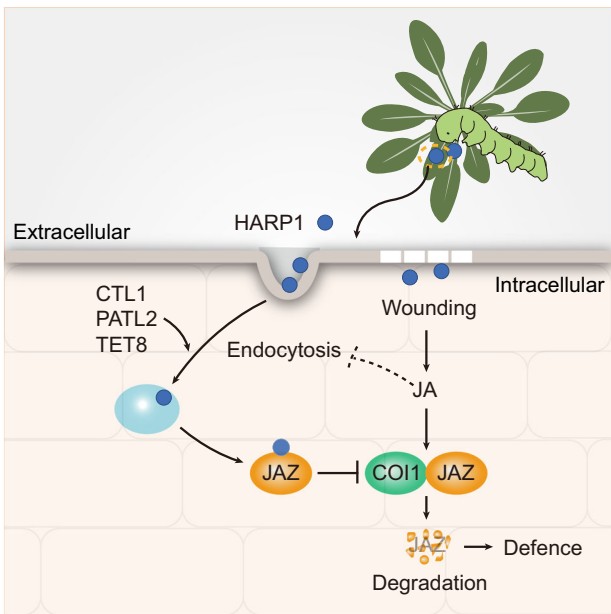

**Fig. 7 | Diagram of HARP1 trafficking in plant.** HARP1 is secreted from oral secretion of cotton bollworm and immigrates into plant cells. Its import is also mediated by CTL1, PATL2 and TET8 related endocytosis and vesicle trafficking. After getting into cells, HARP1 interacts with JAZs to suppress JA signaling output. On the other hand, wounding-induced JA counters HARP1 import by inhibiting endocytosis and HARP1 loading to endosomes setting up a counter-defense loop. Dash line indicates that the mechanism of JA on restriction of endocytosis requires further study.

## Prokaryotic expression and purification of tested proteins

For His-fused proteins (Venus, V-HARP1, V-HARP1δC5, V-HARP1δN34, V-HARP1δN39, V-HARP1δN44, V-HARP1δN49, dsRed-HARP1, mCherry-HARP1), the ORF of the indicated genes were inserted into pETDuet-1 vector with a His N-terminal fusion. For GST-fused proteins (GST-CTL1EC1, GST-PATL2C110, GST-TET8EC2), the ORF of the indicated genes were inserted into pGEX−4T-1 vector with a GST N-terminal fusion. The primers used for the vector constructions are given in Supplementary Date files 1.

The constructed vectors were transformed into *Escherichia coli* strain BL21 (DE3, WeiDi) strain. 0.25 mM isopropyl β-D-thiogalactopyranoside (IPTG) was used to induce protein expression. *E. coli* cells carrying the designed vectors were cultured in 37 °C. After IPTG addition, the cells were transferred to 16 °C for another 12–16 h before harvested. The His fusion proteins were purified by Ni affinity column (Ni-NTA resin, Qiagen) and the GST fusion proteins were purified by Glutathione Sepharose 4B resin (GE Healthcare). The eluted proteins were then concentrated and desalted using an Amicon Ultra-15 Centrifugal Filter Unit (10,000 molecular weight cutoff [MWCO], EMD Millipore) with 50 mM Tris−HCl (pH = 8.0) to a final protein concentration of 1 mg/ml.

## Transient expression assay in Nicotiana benthamiana (N. benthamiana)

About 4–5 monoclones of *Agrobacterium* (GV3101, WeiDi) carrying the target genes were grown together overnight in liquid LB medium under 30 °C, then were transferred to the new liquid LB medium at a ratio of 1:500 and grown overnight. The cultures were centrifuged at room temperature and cells were resuspended with infiltration buffer (10 mM MgCl₂, 10 mM MES, 150 μM Acetosyringone, pH = 5.7) at an OD₆₀₀ of about 1.0. The cell solutions were infiltrated into the back of tobacco leaves. Two to three days later, leaves were harvested for BiLC and pull-down assay.

## Bimolecular luciferase complementary (BiLC) assay

For HARP1 interaction with CTL1 and PATLs (PATL1,2,3), CTL1, PATL1, PATL2 and PATL3 were fused to the amino-terminal half of LUCIFER-ASE (nLUC), HARP1 was fused to the carboxyl-terminal half of LUC (cLUC). For HARP1 interaction with TET8, HARP1 was fused to nLUC and TET8 was fused to cLUC. cLUC and nLUC alone were used as controls. The oligonucleotide primers used for these vectors are given in Supplementary Data file 1. *Agrobacterium* (GV3101, WeiDi) cell solutions were infiltrated into tobacco leaves. Luciferin (1 mM) was infiltrated before LUC activity was monitored after 2 days.

## OS collections and preparations

The fourth and fifth instar larvae were fed with wild-type Arabidopsis (Clo-0) for 1 day. Oral secretions (OS) were than collected[36]. The larva was gently immobilized between the thumb and forefinger, followed by gentle stimulation of the larval mouthpart using a 10 μL pipette tip to induce saliva secretion. OS was subsequently transferred into tubes followed by centrifugation at 12,000 rpm for 10 min at 4 °C. The supernatants were used for further assays.

## Immunoblot and whole amount immunohistochemistry

To detect whether HARP1 import into plant leaves would be inhibited by endocytosis. The wounded *Arabidopsis* leaves were incubated with 50 mM Tris-HCl containing 30 μM A23 and equal amount of DMSO (mock treatment) respectively. After half an hour, A23-pretreated leaves were divided into two groups and transferred independently to the V-HAPR1 solution (1 mg/ml V-HARP1 diluted with 50 mM Tris-HCl to a final concentration of 0.01 mg/ml) supplemented with 30 μM A23 and to the 50% OS (diluted with 50 mM Tris-HCl) supplemented with 30 μM A23. As parallel control, DMSO-pretreated leaves were divided into two groups and transferred independently to the V-HAPR1 solution supplemented with equal amount of DMSO and to the 50% OS supplemented with equal amount of DMSO. After additional 2-h incubation, leaves were then washed for 10 min with 50 mM Tris-HCl, repeated for five times. Samples were than ready for immunoblot and whole amount immunohistochemistry analysis.

For immunoblot assay, the total protein extractions were extracted. Anti-HARP1 antibody[36](dilution, 1:1000) was used for HARP1 detection. For whole amount immunohistochemistry, samples were immediately fixed in FAA buffer (50% methyl alcohol, 5% glacial acetic acid, 3.7% methanal, 41.3% ddH₂O). The fixed samples were dehydrated through a series of graded alcohol solutions and followed by rehydration. After incubation for 30 min with 0.3% H₂O₂, the leaves were transferred to blocking buffer (PBS containing 0.1% Tween 20 and 1% BSA) for an additional 2 h. Then the anti-HARP1 antibody[36] (1:200 dilution) was added. After incubation in 4 °C overnight, the leaves were washed with PBST (PBS containing 0.1% Tween 20) for 10 min and repeated for six times. The HARP1 signals were visualized by Western Blue stabilized substrate for Alkaline Phosphatase and observed under an Olympus microscope.

## Pull-down assay

Full-length V-HARP1 were expressed in *N. benthamiana* leaves. V-HARP1 and Venus were inserted into pCAMBIA1300 vector with a flag C-terminal fusion and then transiently expressed in *N. benthamiana*. The soluble proteins of *N. benthamiana* leaves transiently expressing V-HARP1-flag and Venus-flag (control) were extracted in extraction buffer (50 mM HEPES pH = 7.5, 50 mM NaCl, 10 mM EDTA, 10% glycerol, 0.2% Triton X-100, 2 mM DTT, 1 mM phenylmethylsulfonyl fluoride, 50 μM MG-132 [Targetmol, USA], and 500 × protease inhibitor cocktail). His-fused V-HARP1δN39, V-HARP1δN44 were prokaryotically expressed and purified. About 10 μg proteins of prokaryotically expressed GST, GST-CTL1EC1, GST-PATL2C110 and GST-TET8EC2 were added 20 μL Glutathione Sepharose 4B resins respectively, and then incubated with total protein extracts (about 250 μg) of *N. benthamiana*

leaves in 1 mL extraction buffer or His-fused V-HARP1δN39, V-HARP1δN44 (about 10 µg) in 1 mL 50 mM Tris–HCl (pH = 8.0) at 4°C with rotation. Then, the resins were washed with wash buffer (50 mM HEPES pH 7.5, 150 mM NaCl, 10 mM EDTA, 10% glycerol, 0.1% Triton X-100) for several times. Anti-Flag antibody (ABclonal; dilution, 1:2500) was used to detect V-HARP1-flag and Venus-flag or anti-His antibody (ABclonal; dilution, 1:2500) was used to detect V-HARP1δN39 and V-HARP1δN44 in input and output. Anti-GST (ABclonal; dilution, 1:2500) antibody was used to detect GST, GST-CTL1EC1, GST-PATL2C110 and GST-TET8EC2 in input.

## Yeast two hybrid

V-HARP1 variants were introduced into the pGBKT7 (Clontech). CTL1EC1, PATL2C110 and TET8EC2 were introduced into the pGADT7 (Clontech). The oligonucleotide primers used for these vectors are given in Supplementary Data file 1. A LiCl polyethylene glycol method was used to transfer the indicated plasmids into yeast strain AH109 (Clontech). Transformants were grown on SD-Leu-Trp mediums for 2–3 days and then tested on SD-Leu-Trp-His mediums (-L-T-H) or SD-Ade-Leu-Trp-His mediums (-A- L-T-H) with the indicated 3-amino-1,2,4 triazole (3-AT). At least ten individual clones for each transformant were analyzed to confirm the interactions.

## RNA-seq and transcriptome analysis

The leaves of 10-day-old wild type (Col-0) and JA synthesis mutant (*aos*) were wounded (W) and harvested 2 h post wounding. The untreated plant leaves (CK) were used as control. Total RNA was extracted using the Plant RNA purification kit (Qiagen, 74904). Library construction and RNA-sequencing with three biological replicates was performed on an Illumina HiSeqXten platform (Illumina, San Diego, CA, United States) at Majorbio (Shanghai, China). The clean reads were mapped to the *Arabidopsis* genome (TAIR10) using Hisat2 v2.0.4. HTSeq v0.9.1 to count the reads numbers mapped to each gene. Fragments Per Kilobase of exon model per Million mapped fragments (FPKM) was calculated based on the gene length and the mapped-reads counts. Genes with a fold-change > 2 and FDR < 0.05 were considered as differentially expressed genes (DEGs) using the R package DESeq2[78]. Gene Ontology (GO) enrichment and network analysis were performed by using the Cytoscape v3.8.2 plug-in ClueGO[79]. Significant enrichment was defined as FDR <0.05. GO clusters were inferred by Kappa score. The complete list can be found in Supplementary Data file 2. The raw data are deposited in the NCBI (BioProject accession number: PRJNA760932) and will be released as soon as publication. The reviewer link: https://dataview.ncbi.nlm.nih.gov/object/PRJNA760932?reviewer=kbk81o7uvur0kpn4dh4kpus2lr.

## Gene expression analyses

Plant total RNA was extracted by Trizol reagent (Invitrogen). 1.5 ug of total RNAs were treated with DNase I (1 unit per ul; Fermentas) and used for cDNA synthesis with oligo (dT) primer (TransGen Biotech). qRT-PCR was performed using SYBR green PCR master mix (TaKaRa) on a real-time PCR system (CFX thermocycler; Bio-Rad, Hercules, CA). S18 in *Arabidopsis* (At4g09800) was used as an internal standard. The gene average expression levels were calculated from $2^{-\Delta\Delta Ct}$ values. At least three biological triplicates with technical triplicates were performed. The oligonucleotide primers for all the genes tested are given in Supplementary Data 1.

## Statistical analysis

Data are presented as mean ± SEM. Significances were examined by two-sided Student's *t*-test or by one-way or two-way ANOVA followed by multiple comparison tests with GraphPad Prism software. No less than three independent experiments were performed for each assay, and every experiment contains at least three biological replicates.

## Reporting summary

Further information on research design is available in the Nature Portfolio Reporting Summary linked to this article.

## Data availability

The RNA-seq data that support the findings of this study have been deposited in the National Center for Biotechnology Information (NCBI) under the following accession codes: PRJNA760932. The relevant raw data from each figure are provided in the Source Data file. All data supporting the findings of this study are included in this article, Supplementary Information and Source Data file. Source data are provided with this paper.

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

## Acknowledgements

We thank Yi-Qun Gao for VHA-a1-mRFP, mCherry-CLC2, Rha1-mCherry, *pCTL1::CTL1-GFP* and *ctl1* seeds; Ji-Rong Huang for PIP2A-RFP seeds; Chuan-You Li for *aos* seeds; Gregg A. Howe for *jazQ* seeds; Dao-Xin Xie for *coi1-2* seeds; Jia-Wei Wang and Zu-Hua He for helpful discussions and suggestions; Shui-Ning Yin for confocal microscopy assistance. This research was supported by National Natural Sciences of China Grants (32072430, 31772177, 31788103 to Y.-B.M. and 32000221 to M.-Y.W.); The Ministry of Agriculture of China Grant 2016ZX08009001-009 to Y.-B.M.; The Ministry of Science and Technology (2016YFA0500800 to Y.-B.M.); and the Strategic Priority Research Program of the Chinese Academy of Sciences Grant (XDB11030000 to Y.-B.M.).

## Author contributions

Z.-W.Y., F.-Y.C. and Y.-B.M. designed the research. Z.-W.Y. and F.-Y.C. performed the main experiments with the assistances from X.Z., W.-J.C., C.-Y.C., J.L., M.-N.W., N.-J.L., B.M. and M.-Y.W. D.-Y.C., C.-J.G. and Y.-B.M. analyzed the data. The manuscript was written by Y.-B.M., Z.-W.Y. and F.-Y.C. with input from all authors.

## Competing interests

The authors declare no competing interests.
