## [Peer Review File · Nature Communications]

Endocytosis-mediated entry of a caterpillar effector into plants is countered by JasmonateREVIEWER COMMENTS

Reviewer #1 (Remarks to the Author):

The manuscript describes potential import of the caterpillar effector protein HARP1 into plant cells. While this is potentially an exciting finding, all conclusions regarding localization rely on Venus tagged HARP1 protein applied to wounds at artificial concentrations. In addition, critical control experiments are not presented, as I describe below. I find the conclusions to be poorly supported as a result.

In Fig. 2, the inhibitors A23 and Wm reduce visual amount of Venus signal at the wound site. However, subcellular localization is not convincingly shown. Quantifying number of puncta, not just co-localization with FM-64, should be shown.

In Fig. 3, import in mutant lines is tested, but puncta formation is not measured. Experiments similar to Suppl. Fig 5 should be performed, where the protein is injected. This would remove diffusion into the wound as a confounding factor.

In Fig. 4, truncated HARP1 proteins are tested for uptake, but again subcellular localization / puncta formation is not measured. Again, experiments similar to Suppl. Fig 5 should be performed, where the protein is injected. Point mutants would be much more convincing than truncations. Is there a potential uptake motif shared among effector proteins?

Finally, the conclusion that JA counteracts effector uptake is also not well supported. JA mutant lines could be affected in other growth or developmental processes which influence endocytosis more generally. For the JA mutant line experiments, puncta counts data are quantified in Fig 6c, but this same analysis was not performed with other critical experiments (Figs 2-4)

There are many small issues with language throughout the manuscript. An English-language editing service would increase the readability.

Reviewer #2 (Remarks to the Author):

Manuscript by Chen and co-authors present an interesting insight into the internalization of HARP1 protein extruded from the cotton bollworm. By performing life cell imaging, electro microscopy and biochemistry, they showed that HARP1 is being internalized into the leaf pavement cells by active transport using association with several trafficking proteins. Upon internalization, HARP1 interacts with JA signaling to interfere with host defense. Interestingly, JA can block endocytosis and prevent internalization of HARP1. Overall, the language of the manuscript should be improved, as the structure of the sentences did not always make complete sense.

Several comments for the authors:

1) I am not sure I understand the HARP1 mode of action that authors are trying to propose in the manuscript. In several places they mentioned that HARP1 is being “loaded on the endosomes”, which to me evoke the association from outside. This is not going together with the HARP1 interaction with other endocytic machinery as CTL1, PAT2 or TET8. If HARP1 is

interacting with extracellular parts of these proteins, during the formation of the vesicle it will end up inside of the vesicle. This will lead to the HARP1 getting inside of TGN and PVC/MVB as observed in Fig. 1, which will result of HARP1 vacuolar targeting and degradation. How exactly is HARP1 suppose to escape from the inside of the endosomal compartments so that it can interact with the JA signaling?

2) Visualization and localization of HARP1 is somehow strange. Big disadvantage is that the image of close-up localization of HARP1 around the wound is buried down as Supplementary Figure 18. I would suggest to include more of the images of such magnification. Further, authors stated that HARP1 creates granules, however they never really explain or show properly why exactly these structures are supposed to be called granules. I would imagine would imagine a cluster of a protein, but there is nothing like this present in any of the images in Fig. 1. Moreover, TEM images also did not support this claim as the immunogold labelling rather shows single particles instead of clusters of labelling (which could represent a granule). Why the TEM exhibits only so sparse labelling compared to the high fluorescent signal intensity present in the confocal imaging? There is also no intracellular labelling in the TEM images.

3) Why did authors specifically focused on investigating *ctl1*, *sld1* *sld2*, *pat2*, *tet8*, *PDLP5*, *pat1* and *pat3*, but not the other proteins involved in vesicle trafficking? There is no explanation provided. What about other members of the proteins from the same protein families? Does HARP1 interact with some other proteins as well, or just with these? *CTL1*, *PAT2* and *TET8* showed the most prominent interaction with HARP1 during its internalization. Would it be possible to inhibit HARP1 endocytosis by creating *ctl1/pat2/tet8* triple mutant?

4) Why is that worm feeding experiment divided into three independent parts with some being performed for 3 days and some for 4 days? Like this it is unable for us to compare the results between each other.

5) Is there a negative control present in the Y2H experiment? Authors could show that the other shorter EC loops of these proteins are not interacting with HARP1, or authors could use EC loops of *PATL1* or *PATL3*.

6) The input gel of the Co-IP in Fig. 4a looks strange (especially compared to the one in 4b). Authors should improve that.

7) Also, why is the interaction of HARP1 with *TET8EC2* producing green yeast? Can the image quality be increased?

8) Line 87 – “Confocal microscopy observation shows...” the sentence is missing reference to either publication or figure.

9) Line 99 – Why did authors used rice callous instead of the leaves as for the other species?

Reviewer #3 (Remarks to the Author):

This work extends the breakthrough discovery of Chen et al. 2019 in PNAS on HARP1, (an effector from the OS of *Helicoverpa armigera* that binds to JAZ co-receptor proteins in the nucleus of attacked cells and suppresses JA-induced defense responses) by examining how this effector is imported into the nucleus.

The authors provide compelling evidence that HARP1 is transported to the nucleus by endosome-mediated vesicle trafficking mediated by the vesicle trafficking related proteins, CTL1, PATL2 and TET8 and that endocytosis is the means by which HARP1 is imported into the cell. HARP1-VENUS fusions granulate at the wound-site when applied at high concentrations, and appears to move at a rate and manner, consistent with their import being trafficked by vesicles. Confocal co-localization of the VENUS signal with FM4-64 stained endosomes is again consistent with the proposed mechanism. Pretreatment with the endocytosis inhibitors, A23 and Wm, decreases the fluorescent signal, consistent with an endocytotic mechanism. A series of protein-protein interaction assays (Y2H, pulldowns, and BiLC assays) provide evidence that CTL1, PATL2 and TET8 directly interact with HARP1. Deletions of the C-terminal 5aa and 44 aa of the N-terminal of HARP1 abolished these interactions. In summary, this work establishes that this insect effector is vectored into plant cells by similar mechanisms to those which vector fungal effectors into cells, even if the precise HARP1 residues that interact with the trafficking proteins could not be identified in this study. The potential alternative mechanisms (diffusion, active transport, plasmodesmata transport) are unlikely, given the physical characteristics of HARP1.

The authors then provide evidence using the same confocal toolbox in combination with different JA-mutants, (JA-biosynthesis: aos and two co-receptor mutants: JazQ and COI1) that JA-signaling inhibits endocytosis (only with aos) and that JA inhibits HARP1 loading onto endosomes (with all three mutants). For this part of the story, I missed the discussion that three JA mutants used by the authors will likely have very different levels of JAZ proteins at the wound site, likely in the following order: AOS > COI1 > COL-O > JAZQ, which given that HARP1 binds to some of the JAZ family members (as reported in the authors' previous work), should be considered in interpreting the endosome signals.

The packaging of these findings, that JA-suppression of the endocytotic process, as an evolutionary counter-response to the insects' evolution of JA-suppressive HARP1, seems premature given the evidence provided, and will likely require a deeper functional analysis than is provided here for this to be a reasonable inference.

In an attempt to establish the functional relevance of the import mechanism for the plant-insect interaction, the authors evaluate the influence on larval weight gain and wound-elicited transcriptional responses of putative defense genes. Wound-induced transcripts are reduced by V-HARP1 pretreatment in WT plants, but not as strongly in the vesicle trafficking-related mutants, *ctl1*, *patl2* and *tet8*, consistent with a modest attenuation of induced defense responses. Larval weight gain was very modestly reduced when feeding on *ctl1* and *patl2* mutants, but not on *tet8* mutant plants. The differences among the mutants is not considered and needs to be discussed. Why was there no effect in the *tet8* line?

The very modest effect of blocking the import mechanism on larval weight gain is perhaps not surprising given the very modest decrease in weight gain of larvae silenced in HARP1 expression (by dsRNAi), and the very modest suppressive effects on JA-induced defense-related transcripts mediated by HARP1, as published in the Chen et al paper. In other words, the JA-signaling suppressive effects of HARP1 are not very strong. It would have been a valuable benchmark, had the authors' included the defenseless aos mutant in their larval

mass analysis; this would have allowed the functional effect size to have been calibrated against the likely very large weight gain seen in larvae feeding on aos plants.

The lack of important details in the insect bioassays make it difficult to evaluate the significance of these very modest effects and impoverishes the discussion of the data.

Third instar larvae were used in the assays. What was the diet of the bollworms prior to being used in the feeding assays? This is an important detail to include, as the authors' previous publication demonstrated that diet has a dramatic effect on larval OS HARP1 concentrations. Had the larvae been reared for their first two instars on gossypol-containing diets or the non-preferred Arabidopsis plants, their HARP1 concentrations would have been very high, and one would have expected to have seen much larger effects in the larval weight gain assays.

In the assays, purified HARP1 was used, and at concentrations likely 1000-fold higher than one would expect to have been transferred to a feeding site in an *H. armigera*-*Arabidopsis* interaction (a non-preferred hostplant). Moreover, the authors' "counter-defense response" interpretation of the function of JA-signaling's apparent inhibition of HARP1 import through the inhibition of endocytosis and HARP1 loading onto endosomes is poorly tested by the sole use of purified HARP1. Had the authors used intact OS and HARP1-free OS, their analysis would have provided a more convincing case for the "counter-evolutionary" response interpretation. This is because OS contains other elicitors, such as FACs, which are known to dramatically amplify the JA-burst that is elicited when bollworms feed on preferred hosts. And the interesting question would lie in the outcome of these antagonistic processes: is the FAC-elicited JA-burst sufficient to override the JA-elicited inhibition of HARP1 importation and HARP1's inhibitory effect on JA signaling? HARP1 concentration are clearly rapidly responsive to diet; are FAC and HARP1 concentrations in OS inversely related?

Clearly these additional questions are out of the scope of this short communication, and hence without stronger data and more thoroughly described techniques, the counter-defense response interpretation should be given a more nuanced presentation, if not removed.

Reviewer #4 (Remarks to the Author):

The authors investigated endocytosis of the HARP1 protein of *Helicoverpa armigera* into plant cells, primarily using *Arabidopsis* as a model system. Several independent lines of evidence indicate that HARP1 is taken up by endocytosis, including: 1) direct interactions with CTL1, PATL2 and TET8 in pull-down experiments, 2) co-localization in plant cells, 3) lack of uptake when endocytosis is chemically inhibited, 4) analysis of plant mutants, and 5) experiments with truncated HARP1 protein. Interestingly, jasmonic acid elicitation has a negative effect on HARP1 endocytosis, suggesting that the plant is actively countering HARP1-mediated defense suppression.

Overall, the data that are presented support the main conclusions of the manuscript. These main conclusions (endocytosis of HARP1 into plant cells and suppression of this process by jasmonic acid) are a significant contribution to research on plant-insect interactions.

Specific comments:

Line 91: 0.1 mg/ml HARP1 is used for the experiments. Is this a physiologically relevant concentration? It would be good if the authors commented on the concentration of HARP1 in caterpillar spit relative to what they used in their experiments.

The manuscript contains numerous small English-language errors that need to be corrected.

Answers and Responses

I agree that most of the suggestions by the reviewers are helpful to further improve our manuscript. Accordingly, we have made a substantial revision. All the main modifications were highlighted in the main text file.

Major Revisions:

1, Observations of injected HARP1 in WT, *ctl1*, *patl2* and *tet8* by confocal microscopy (Fig.3c).

When purified recombinant HARP1 was injected into leaves of the wild type (Col-0), *ctl1*, *patl2* and *tet8*, HARP1 was tended to cluster as granules in wild type and the granulated V-HARP1 was largely reduced in *ctl1*, *patl2* and *tet8*. This result indicated that the vesicle trafficking related proteins CTL1, PATL2 and TET8 were responsible for HARP1 granulation in plant leaves.

2, Observations of V-HARP1 δ N39 and V-HARP1 δ N44 after injected into plant leaves (Supplementary Fig. 14e, f).

In Fig.4f, V-HARP1 δ N39 could still enter into plant cells successfully while V-HARP1 δ N44 could not. Interestingly, when V-HARP1 δ N39 and V-HARP1 δ N44 were injected into plant leaves, it was V-HARP1 δ N39 but not V-HARP1 δ N44 was observed as granules (Supplementary Fig. 14e). This suggested that the granulated characteristics of HARP1 were correlated to its successful import into plant cells.

3, Analysis whether HARP1 import into *jar1* was affected by the presence of MeJA (Supplementary Fig. 19a-c).

In Figure 6, we analyze whether HARP1 import was affected in the JA synthesis mutant *aos*, JA insensitive mutant *coi1* and JA hypersensitive mutant *jazQ*. To further conform the JA effect on HARP1 import, in this revision, we added the analysis of another JA synthesis mutant *jar1*. JA-Ile is the active molecule of the JA signaling and JAR1 catalyzes the conjugation of JA with isoleucine to produce JA-Ile. In *jar1*, the deficiency of generating JA-Ile from JA caused insensitive response to JA treatment. As shown in Supplementary Fig. 19a-c, both the HARP1 entry and endosome accumulations in *jar1* were not affected by the presence of MeJA. Combined with the observations from Figure 6, it suggested that JA restricted HARP1 import by inhibiting plant endocytosis.

4, The manuscript was carefully examined and the grammar and spelling errors were collected.

Response to reviewers:

Response to the reviewer #1

Comment 1: In Fig. 2, the inhibitors A23 and Wm reduce visual amount of Venus signal at the wound site. However, subcellular localization is not convincingly shown. Quantifying number of puncta, not just co-localization with FM-64, should be shown.

Response 1: Thank you for the comment. We agree that quantification of endosomes and HARP1 granules after different endocytosis inhibitor treatments would provide more information. However, a part of endosomes were aggregated by BFA treatment and it is hard to distinguish that how many

granules were aggregated in the BFA body. On the other hand, the FM4-64 traced endosomes is
extremely rare after A23 and Wm treatment which can be seen in Fig. 2 and in the below image
(large view). Although we didn't calculate the number of the granules, the quantified signal intensity
of HARP1 was given in Fig. 2b. It revealed that HARP1 import was largely inhibited by A23 and Wm
treatment.

Confocal microscopy observation of V-HARP1 in A23 (a) and Wm (b) pretreated leaves. Bar: 10 μ m

**Comment 2:** In Fig. 3, import in mutant lines is tested, but puncta formation is not measured.
Experiments similar to Suppl. Fig 5 should be performed, where the protein is injected. This would
remove diffusion into the wound as a confounding factor.

**Response 2:** Thank you for your suggestion. In this revision, we provided extra data of the
observations of HARP1 after it was injected into WT, *ctl1*, *patl2* and *tet8* by confocal microscopy. As
to the results, HARP1 was tended to cluster as granules in wild type and the granulated V-HARP1
was largely reduced in *ctl1*, *patl2* and *tet8*. This result indicated that the vesicle trafficking related
proteins CTL1, PATL2 and TET8 were responsible for HARP1 granulation in plant leaves.

**Comment 3:** In Fig. 4, truncated HARP1 proteins are tested for uptake, but again subcellular
localization/ puncta formation is not measured. Again, experiments similar to Suppl. Fig 5 should be
performed, where the protein is injected. Point mutants would be much more convincing than
truncations. Is there a potential uptake motif shared among effector proteins?

**Response 3:** Thank you for your suggestion. In this revision, we provided the data about the
observation of truncated HARP1 proteins after injected into plant leaves by confocal microscopy. In
Fig.4f, V-HARP1 δ N39 could still enter into plant cells successfully while V-HARP1 δ N44 could not.
Interestingly, when V-HARP1 δ N39 and V-HARP1 δ N44 were injected into plant leaves, it was
V-HARP1 δ N39 but not V-HARP1 δ N44 was observed as granules (Supplementary Fig. 14e). This

suggested that the granulated characteristics of HARP1 were correlated to its successful import into
plant cells.

Phylogenetic analyses of proteins in nine species of Lepidoptera that have significant homology to
the venom R-like proteins (AGM32454.1) in *Coptotermes formosanus* were performed. These results
can be found in our recently reported paper (Chen et al. 2023, New Phytologist). However, the
functions of these homology proteins need to be identified. I think when we have a better
understanding of the function of these proteins and maybe it would be possible to find out the
potential uptake motif. Thank you for your comment.

**Comment 4:** Finally, the conclusion that JA counteracts effector uptake is also not well supported. JA
mutant lines could be affected in other growth or developmental processes which influence
endocytosis more generally. For the JA mutant line experiments, puncta counts data are quantified in
Fig 6c, but this same analysis was not performed with other critical experiments (Figs 2-4).

**Response 4:** We agree with you that JA could affect the growth and developmental processes. In
Fig.6, the plant leaves were transiently treated with MeJA (2 hours) and this largely reduced the JA
effects on plant development. And in this revision, we added the analysis of another JA synthesis
mutant *jar1*. JAR1 catalyzes the conjugation of JA with isoleucine to produce JA-Ile which is the
active molecule in JA signaling pathway. In *jar1*, the deficiency of generating JA-Ile from JA caused
insensitive response to JA treatment. As shown in Supplementary Fig. 19a-c, both the HARP1 entry
and endosome accumulations in *jar1* were not obviously affected by the presence of MeJA.
Combined with the observations from Figure 6, it suggested that JA restricted HARP1 import by
inhibiting plant endocytosis. We agree with you that further studies are required to demonstrate the
mechanism of JA regulation on endocytosis and that would be another interesting story. Thank you
for your comment.

**Comment 5:** There are many small issues with language throughout the manuscript. An
English-language editing service would increase the readability.

**Response 5:** Thank you! The manuscript was carefully examined and the grammar and spelling
errors were collected.

**Response to the reviewer #2**

**Comment 1:** I am not sure I understand the HARP1 mode of action that authors are trying to
propose in the manuscript. In several places they mentioned that HARP1 is being “loaded on the
endosomes”, which to me evoke the association from outside. This is not going together with the
HARP1 interaction with other endocytic machinery as CTL1, PAT2 or TET8. If HARP1 is interacting
with extracellular parts of these proteins, during the formation of the vesicle it will end up inside of the
vesicle. This will lead to the HARP1 getting inside of TGN and PVC/MVB as observed in Fig. 1, which
will result of HARP1 vacuolar targeting and degradation. How exactly is HARP1 suppose to escape
from the inside of the endosomal compartments so that it can interact with the JA signaling?

**Response 1:** CTL1 and TET8 are transmembrane proteins, their outside membrane domains
(CTL1EC1 and TET8EC2) could interact with HARP1 (Fig. 4). In our opinion, HARP1 enters plant
cells via CTL1, PAT2 and TET8 mediated endocytosis. The statement of “loaded on the endosomes”
was inappropriate. We have modified these descriptions as “loaded into the endosomes”. Thank you
for your comment.

After getting inside of the plant cells, HARP1 was observed colocalizing with the nucleus and
chloroplasts. In animal systems, endosome escape is triggered by acidification of the endosomal
lumen. Concanamycin A (ConcA) is an inhibitor of endosome acidification. We then use ConcA to
test whether HARP1 escapes from endosomes by the acidification and reaches chloroplasts
triggered. As shown in the following figure, when plants were treated with ConcA, HARP1 cannot
colocalize with chloroplast anymore.

**Z-stack scanning of V-HARP1 colocalized with chloroplast.** The wounded leaves of wild type
were incubated with V-HARP1 in the presence of 1 μ M ConcA (b) or in the presence of equal volume
of DMSO (control, CK) (a). Chloroplast was detected by auto fluorescence. The lines indicated the
vertical and horizontal cross-sections shown at the right and bottom respectively. Arrows indicate the
orientation from pavement to mesophyll cells. Scale bar: 10 μ m.

This study was focus on HARP1 transportation from out to inside plant cells. We agree with you that
how HARP1 reached the target organelles after transported into cells is a very interesting question
and deserves further study. Also it should be another story, so the above figure was not included in
this manuscript. I sincerely hope to have a deeper discussion with you when we have more data.

**Comment 2:** Visualization and localization of HARP1 is somehow strange. Big disadvantage is that
the image of close-up localization of HARP1 around the wound is buried down as Supplementary
Figure 18. I would suggest to include more of the images of such magnification. Further, authors
stated that HARP1 creates granules, however they never really explain or show properly why exactly
these structures are supposed to be called granules. I would imagine would imagine a cluster of a
protein, but there is nothing like this present in any of the images in Fig. 1. Moreover, TEM images
also did not support this claim as the immunogold labelling rather shows single particles instead of
clusters of labelling (which could represent a granule). Why the TEM exhibits only so sparse labeling

compared to the high fluorescent signal intensity present in the confocal imaging? There is also no
intracellular labelling in the TEM images.

**Response 2:** Cells around the wound were seriously injured and the cell states were varied. In
addition, the signal intensities are too strong to be observed clearly. So we selected the region with
relatively clear signals and stable cell states for observation.

We think that HARP1 granules were caused by its colocalization with endosomes. After endocytosis
was inhibited by A23 and Wm, the accumulations of endosomes were dramatically reduced to a very
low level (Fig. 2a, c) and accompanied with the decrease of granulated V-HARP1. In this revision, we
provided new data about the observations of V-HARP1 δ N39 and V-HARP1 δ N44 after injected into
plant leaves (Supplementary Fig. 14e). In Fig.4b, V-HARP1 δ N39 could interact with the vesicle
trafficking related proteins CTL1, PATL2 and TET8 and enter into plant cells successfully while
V-HARP1 δ N44 could not. Interestingly, when V-HARP1 δ N39 and V-HARP1 δ N44 were injected into
plant leaves, it was V-HARP1 δ N39 but not V-HARP1 δ N44 was observed as granules
(Supplementary Fig. 14e). This result also supported the conclusion that HARP1 granules were
caused by its colocalization with endosomes.

We agree with you that the HARP1 observed from confocal microscopy and TEM are different. In
confocal microscopy, HARP1 was visualized by fusing to Venus. And in TEM assay, the positive
signals were the immunogold labeled HARP1. Therefore the variation might be caused by different
detections. Similar phenomenon was also found in other research papers. For your convenience,
here, we selected a similar case from the research article of Nature Plants (Li et al. 2021). The
following image is comprised of Fig. 1d and Fig. 2a from the research article (Li et al. Nature Plants 7,
2021). Please note the blue circles marked regions which displayed the immunogold labeled Man1
signals in TEM assay (Fig. 1d) or visualized by fusing Man1 to RFP in confocal image (Fig. 2a)

**This image is comprised of Fig. 1d and Fig. 2a from the research article (Li et al. Nature Plants**
**7, 2021). Fig. 1d,** TEM analysis of the negatively stained vesicles with immune-gold labeling by
COPI/II coat or cargo proteins. Scale bars, 100 nm. **Fig.2a,** Confocal microscopy images of
Man1-RFP ER export co-expressing AtSar1a/ cDN-GFP in protoplast. Scale bars, 10 μ m.

**Comment 3:** Why did authors specifically focused on investigating *ctl1*, *sld1* *sld2*, *patl2*, *tet8*, *PDLP5*,
*patl1* and *patl3*, but not the other proteins involved in vesicle trafficking? There is no explanation
provided. What about other members of the proteins from the same protein families? Does HARP1
interact with some other proteins as well, or just with these? CTL1, PAT2 and TET8 showed the most
prominent interaction with HARP1 during its internalization. Would it be possible to inhibit HARP1
endocytosis by creating *ctl1/pat2/tet8* triple mutant?

**Response 3:** Considering that HARP1 is imported through endocytosis, we questioned whether its
import would be affected in endocytosis related mutants. CTL1, Sphingolipids Delta-8 desaturase
(SLD), PATLs and TETs regulate endomembrane systems and are involved in maintaining
membrane lipid homeostasis and vesicle trafficking. 35:*PDL5* displays the reduced permeability of
plasmodesmata. These related plant materials had been reported and can be obtained from other
research groups. So these mutants were used in our assay.

We agree with you that *ctl1/pat2/tet8* triple mutant is helpful for our study. However, both *PATL2* and
*CTL1* are located in chromosome one and this increases difficulties for generating *ctl1/pat2/tet8* triple
mutant.

**Comment 4:** Why is that worm feeding experiment divided into three independent parts with some
being performed for 3 days and some for 4 days? Like this it is unable for us to compare the results
between each other.

**Response 4:** In our lab, the larvae of Noctuidae pest were usually used to assay plant resistance, in
detail, 3rd instar larvae were weighted after transferred to the tested plants for 3-4 days. Generally,
3-4 day feeding is sufficient to distinguish the resistances of different plants. The insect feeding test
presented in Fig. S10a and Fig. S10b were performed independently. So comparisons can only be
made between groups within the same experiment. I agree with you, it is better to examine the
resistance of those mutants in the same time. However, with the limited space of Arabidopsis growth
box (Percival), sometimes the insect resistance of all the mutants could not be tested at the same
time. Thank you for your comment.

**Comment 5:** Is there a negative control present in the Y2H experiment? Authors could show that the
other shorter EC loops of these proteins are not interacting with HARP1, or authors could use EC
loops of *PATL1* or *PATL3*.

**Response 5:** Thank you for your comment. I speculated that the intention of this suggestion is to
determine that HARP1 did not interact with *PATL1* or *PATL3*. Though the EC loops of *PATL1* or
*PATL3* were not analyzed in Y2H assay, we had performed BiLC assay to detect HARP1 interaction
with CTL1, *PATL1*, *PATL2*, *PATL3* and TET8, and the results showed that only CTL1, *PATL2* and
TET8 had the interactions with HARP1. Considering that *PATL1* and *PATL3* had no interaction with
HARP1, in the original manuscript, we only provided the results of HARP1 interaction with CTL1,
*PATL2* and TET8. In this revision, the negative BiLC assay results of *PATL1* and *PATL3* were also
provided in order to give the evidence that *PATL1* and *PATL3* could not interact with
HARP1(Supplemental Figure 12c).

**Comment 6:** The input gel of the Co-IP in Fig. 4a looks strange (especially compared to the one
in4b). Authors should improve that.

**Response 6:** Thank you for your comment. We have improved the image.

**Comment 7:** Also, why is the interaction of HARP1 with TET8EC2 producing green yeast? Can the
image quality be increased?

**Response 7:** Thank you for your comment. We have improved the image.

**Comment 8:** Line 87 – “Confocal microscopy observation shows...” the sentence is missing
reference to either publication or figure.

**Response 8:** Thank you for your comment. The reference was added.

**Comment 9:** Line 99– Why did authors used rice callous instead of the leaves as for the other
species?

**Response 9:** Thank you for your comment. When incubated with HARP1 solutions, the surface of
rice leaf was poorly mixed with the protein solutions. We speculated that this might be caused by the
high hydrophobicity of rice leaves. So we used callus instead of leaves.

**Response to the reviewer #3**

**Comment 1:** The authors then provide evidence using the same confocal toolbox in combination
with different JA-mutants, (JA-biosynthesis: aos and two co-receptor mutants: JazQ and COI1) that
JA-signaling inhibits endocytosis (only with aos) and that JA inhibits HARP1 loading onto endosomes
(with all three mutants). For this part of the story, I missed the discussion that three JA mutants used
by the authors will likely have very different levels of JAZ proteins at the wound site, likely in the
following order: AOS> COI1>COL-O>JAZQ, which given that HARP1 binds to some of the JAZ
family members (as reported in the authors ‘previous work), should be considered in interpreting the
endosome signals.

**Response 1:** This is a good question. I agree with you that the JAZ content in these mutants would
likely in the following order: AOS> COI1>COL-O>JAZQ. And with the observation of HARP1 in these
mutants, it seems that JAZ content is also correlated with HARP1 import. We spent quite a long time
to thinking about this and finally came to the conclusion that JAZ was less likely to directly affect
HARP1 import. Following are the two possible reasons:

1, JAZ was mainly located in nucleus, this reduced the possibility to meet HARP1 outside of cells or
in the cytoplasm.

2, In the wounding site, JAZ of the damaged cells might be presented in the cytoplasm or in apoplast.
However, wounding quickly triggered JAZ degradation and JAZ was reduced to a large extent within
15 minutes and was hardly detected 30 minutes post wounding (Chen et al. 2019 PNAS; Mao et al.
2017 Nature Communications). This feature of JAZ minimized its direct impact on HARP1 imports.

We also added some related discussion in this revision (Page10, Line263-257)

**Comment 2:** The packaging of these findings, that JA-suppression of the endocytotic process, as an
evolutionary counter-response to the insects’ evolution of JA-suppressive HARP1, seems premature
given the evidence provided, and will likely require a deeper functional analysis than is provided here

for this to be a reasonable inference.

**Response 2:** In this revision, we added another JA synthesis mutant (*jar1*) for analysis. JAR1
catalyzes the conjugation of JA with isoleucine to produce JA-Ile which is the active molecule in JA
signaling pathway. In *jar1*, the deficiency of generating JA-Ile from JA caused insensitive response to
JA treatment. As shown in Supplementary Fig. 19a-c, both the HARP1 entry and endosome
accumulations in *jar1* were not obviously affected by the presence of MeJA. Combined with the
observations from Figure 6, it suggested that JA restricted HARP1 import by inhibiting plant
endocytosis. We agree with you that further studies are required to demonstrate the mechanism of
JA regulation on endocytosis and that would be another interesting story. In this revision, the
inhibition of JA on endocytosis was indicated with dash which implied that the mechanism of JA on
restriction of endocytosis requires further study.

**Comment 3:** In an attempt to establish the functional relevance of the import mechanism for the
plant-insect interaction, the authors evaluate the influence on larval weight gain and wound-elicited
transcriptional responses of putative defense genes. Wound-induced transcripts are reduced by
V-HARP1 pretreatment in WT plants, but not as strongly in the vesicle trafficking-related mutants,
*ctl1*, *patl2* and *tet8*, consistent with a modest attenuation of induced defense responses. Larval
weight gain was very modestly reduced when feeding on *ctl1* and *patl2* mutants, but not on *tet8*
mutant plants. The differences among the mutants is not considered and needs to be discussed.
Why was there no effect in the *tet8* line?

**Response 3:** CTL1, PATL2 and TET8 regulate endomembrane systems and are involved in
maintaining membrane lipid homeostasis and vesicle trafficking. Besides reduced HARP1 import,
*ctl1*, *patl2* and *tet8* might also differentially influence other aspects of plant. This might be a reason,
to explain why only a mild enhancement on insect resistance was observed in *ctl1* and *patl2* but not
in *tet8*. We have added related discussion in this revision (Page 7 Line164-165).

**Comment 4:** The very modest effect of blocking the import mechanism on larval weight gain is
perhaps not surprising given the very modest decrease in weight gain of larvae silenced in HARP1
expression (by dsRNAi), and the very modest suppressive effects on JA-induced defense-related
transcripts mediated by HARP1, as published in the Chen et al paper. In other words, the
JA-signaling suppressive effects of HARP1 are not very strong. It would have been a valuable
benchmark, had the authors' included the defenseless *aos* mutant in their larval mass analysis; this
would have allowed the functional effect size to have been calibrated against the likely very large
weight gain seen in larvae feeding on *aos* plants.

**Response 4:** Thank you for your comment. We agree with you that JA signaling play predominant
role in plant resistance to insects especially to chewing insects. And we also observed that larvae
reared on *aos* grow quickly in our insect feeding test.

Insect OS secretion has complex ingredients and contains multiple effectors. In our recent research
we identified another effector from cotton bollworm OS which interacted with bHLH transcription
factors including MYC3 and MYC4 in Arabidopsis and inhibited their functions (Chen et al. 2023 New
Phytologist). This gives the evidence of functional redundancy of insect effectors manipulating JA
pathway. We think that the silenced expression of HARP1 only caused mild effects were due to
functional redundancy.

**Comment 5:** The lack of important details in the insect bioassays make it difficult to evaluate the
significance of these very modest effects and impoverishes the discussion of the data.

**Response 5:** In this revision, we make a more detailed method and provide a schematic diagram of
insect feeding test (Page16 Line438-442). Thank you for your comment.

**Comment 6:** Third instar larvae were used in the assays. What was the diet of the bollworms prior to
being used in the feeding assays? This is an important detail to include, as the authors' previous
publication demonstrated that diet has a dramatic effect on larval OS HARP1 concentrations. Had the
larvae been reared for their first two instars on gossypol-containing diets or the non-preferred
Arabidopsis plants, their HARP1 concentrations would have been very high, and one would have
expected to have seen much larger effects in the larval weight gain assays.

**Response 6:** The newly hatched larvae were raised on artificial diet till they were used for assay.
These descriptions had been added in Methods (Page? Line?-?). Thank you! If we use the larvae
which had been reared on Arabidopsis plants, it might increase the significance of the differential
insect resistance between mutants and wild type however, we think the impact would be mild.
Because HARP1 accumulation was induced after the larvae fed on with Arabidopsis for one day,
after the 3rd instar larvae transferred to the tested plants, HARP1 accumulation would be induced
within one day.

Thank you for your comment. Thank you for the helpful advices to perform the insect feeding test
accurately. We think the mild effects on the mutant's resistance to cotton bollworm were caused by
multiple reasons, including:

1, Besides the reduced HARP1 import, *ctf1*, *patl2* and *tet8* might also differentially influence other
aspects of plant.

2, There are multiple effectors in insect OS. HARP1 might have functional redundancy with other
effectors including our recently reported HAS1 (Chen et al. 2023 New Phytologist).

**Comment 7:** In the assays, purified HARP1 was used, and at concentrations likely 1000-fold higher
than one would expect to have been transferred to a feeding site in an *H. armigera*-Arabidopsis
interaction (a non-preferred host plant). Moreover, the authors' "counter-defense response"
interpretation of the function of JA-signaling's apparent inhibition of HARP1 import through the
inhibition of endocytosis and HARP1 loading onto endosomes is poorly tested by the sole use of
purified HARP1. Had the authors used intact OS and HARP1-free OS, their analysis would have
provided a more convincing case for the "counter-evolutionary" response interpretation. This is
because OS contains other elicitors, such as FACs, which are known to dramatically amplify the
JA-burst that is elicited when bollworms feed on preferred hosts. And the interesting question would
lie in the outcome of these antagonistic processes: is the FAC-elicited JA-burst sufficient to override

the JA-elicited inhibition of HARP1 importation and HARP1's inhibitory effect on JA signaling?
HARP1 concentration are clearly rapidly responsive to diet; are FAC and HARP1 concentrations in
OS inversely related?

Clearly these additional questions are out of the scope of this short communication, and hence
without stronger data and more thoroughly described techniques, the counter-defense response
interpretation should be given a more nuanced presentation, if not removed.

**Response 7:** We agree with you that the HARP1 content we used is relatively high. As the insect
effectors, it might have some modifications post the translation which is important for its function.
However, in prokaryotically expression system, the possible modifications might be missing. In our
test, HARP1 was prokaryotically expressed and purified. Heterogeneous expression and subsequent
purification might reduce its protein activity. Also, HARP1 is present in insect oral secretions, which
might be the most fitting "buffer solution" required for HARP1 activity. We think that these might be
the reasons that higher concentration is needed in our test. The related discussions were added in
this revision (Page 11, Line 271-277). Thank you!

In this revision, we added another JA synthesis mutant (*jar1*) for analysis. JAR1 catalyzes the
conjugation of JA with isoleucine to produce JA-Ile which is the active molecule in JA signaling
pathway. In *jar1*, the deficiency of generating JA-Ile from JA caused insensitive response to JA
treatment. As shown in Supplementary Fig. 19a-c, both the HARP1 entry and endosome
accumulations in *jar1* were not obviously affected by the presence of MeJA. Combined with the
observations from Figure 6, it suggested that JA restricted HARP1 import by inhibiting plant
endocytosis. We agree with you that further studies are required to demonstrate the mechanism of
JA regulation on endocytosis and that would be another interesting story.

The impact of OS on plant defense is a comprehensive effect of all the active molecules. The elicitors
like FAC can elicit plant defense while effectors like HARP1 can weaken JA dependent defense. It is
difficult to say, which side would gain the upper hand. And this made the plant-insect interactions
more complex and interesting and full of challenges. Thank you for your comment. Hope we can
discuss further about this interesting question in the future. In this revision, we have added the
introduction of FAC elicitors (Page 3, Line 36-38). Thank you for your suggestion.

**Response to the reviewer #4**

**Comment 1:** Line 91: 0.1 mg/ml HARP1 is used for the experiments. Is this a physiologically relevant
concentration? It would be good if the authors commented on the concentration of HARP1 in
caterpillar spit relative to what they used in their experiments.

**Response 1:** When 0.1 mg/ml V-HARP1 was used for analysis, fluorescence signals could be
detected in cells after incubation although it is still higher than the physiological concentration. As the
insect effectors, it might have some modifications post the translation which is important for its
function. However, in prokaryotically expression system, the possible modifications might be lacking.

In our test, HARP1 was procaryotically expressed and purified. Heterogeneous expression and
subsequent purification might reduce its protein activity. Also, HARP1 is naturally present in insect
oral secretions, which might be the most fitting “buffer solution” required for HARP1 activity. We think
that these might be the reasons that higher concentration is needed in our test. In this revision, we
have added related discussions (Page 11, Line271-277). Thank you for your suggestion.

**Comment 2:** The manuscript contains numerous small English-language errors that need to be
corrected.

**Response 2:** Thank you for your comment. The manuscript was carefully examined and the
grammar and spelling errors were collected.

REVIEWER COMMENTS

Reviewer #1 (Remarks to the Author):

The manuscript addresses my technical concerns, but largely not my larger conceptual concerns. HARP1 injection experiments and granule counts were performed, including with the N39 and N44 truncations. In addition, an experiment with the *jar1* (JA-Ile biosynthetic mutant) was performed with results consistent with other JA mutants and exogenous JA application. I am overall very convinced that exogenously applied HARP1 protein associates with endosomes and that JA reduces this association, although a positive control (for example fungal or oomycete effector) would have been desirable.

However, I am still quite skeptical of the overall conclusion of effector entry during caterpillar attack. While the manuscript provides evidence with purified HARP1 protein, I believe there needs to be some demonstration that HARP1 can be found in endosomes during an actual plant-herbivore interaction.

A recent breakthrough regarding oomycete RXLR effectors (Wang 2023, DOI: 10.1093/plcell/koad069) provides an example approach to demonstrate of HARP1-endosome association during a plant-herbivore interaction. In the paper, the authors used transient expression in *Nicotiana benthamiana* of endosome components NbCLC2-GFP or NbAra6-GFP, followed by oomycete infection. Ultracentrifugation-immunoblot experiments were used to show that RFP-tagged effectors were present in the endosome (GFP) fractions. Since tagged HARP1 caterpillars are not technically feasible, a HARP1 antibody could be developed for this experiment or HARP1 could be detected by mass spectrometry. Detection of native HARP1 in labeled endosomes would greatly alleviate the technical concerns regarding exogenous HARP1 application.

Other comments:

- Line 37-38: The text should be introduce elicitors in general many of which can elicit JA, not just FACs.
- Line 171: Should be "bimolecular" not "bilayer"
- Line 317: Please add accession number XP_047035071 in addition to the URL
- Line 714 and many other places: "sights" should be "sites"

Reviewer #2 (Remarks to the Author):

I would like to thank the authors for addressing the comments.

However, with some of the responses I am not satisfied:

Response 3: The explanation for choosing the candidate genes is rather strange, authors just pick some proteins involved in membrane homeostasis and vesicle trafficking, without clear justification of any biological reasoning for that. It should be stated in the manuscript why these genes we chosen in particular and not the other (even the ones from the same family).

Also authors claim the inability to produce and test the *ctl1/pat2/tet8* triple mutant due to the close proximity of the genes on the first chromosome. This problem can of course be avoided by using the CRISPR technology to mutate the genes (even all three at once...).

Response 4: The feeding experiment should be performed at once using all the possible genotypes and the same conditions (regarding the length of the feeding time). Limited growth space should not be a reason not to perform the experiment in the proper way.

Reviewer #4 (Remarks to the Author):

In the revised manuscript, the authors conducted experiments that addressed most of the reviewer concerns. Some concern that I still have are:

Both I and another reviewer are concerned that the authors used a concentration of HARP1 that is much higher than what normally would be found in caterpillar spit. This is required to see an effect in the authors' assays. Effects that are not physiologically relevant can occur when such a large excess of protein is used for experiments.

The authors say that the high HARP1 concentration was needed because the protein was purified from bacteria and therefore may not have the right insect-specific modifications. I don't think that this is a sufficient response. If that is the hypothesis, then the authors should purify their protein from a lepidopteran cell expression system (e.g., Sf9 or High 5 cells are commonly used for this) and show that there is greater activity than when the protein is purified from bacteria. If the "most fitting buffer solution" of insect oral secretions is further required to obtain activity, then the transgenically produced fusion proteins can be added to insect oral secretions for the assays.

The author response "4, The manuscript was carefully examined and the grammar and spelling errors were collected." does not inspire great confidence in their approach to correcting grammar and spelling errors.

Response to reviewers

We have made a substantial revision according to your suggestions. All the main modifications of this revision were highlighted in blue in order to distinguish the modifications of the first round revision (highlighted in red).

Reviewer #1:

Comment 1: However, I am still quite skeptical of the overall conclusion of effector entry during caterpillar attack. While the manuscript provides evidence with purified HARP1 protein, I believe there needs to be some demonstration that HARP1 can be found in endosomes during an actual plant-herbivore interaction. A recent breakthrough regarding oomycete RXLR effectors (Wang 2023, DOI: 10.1093/plcell/koad069) provides an example approach to demonstrate of HARP1-endosome association during a plant-herbivore interaction. In the paper, the authors used transient expression in *Nicotiana benthamiana* of endosome components NbCLC2-GFP or NbAra6-GFP, followed by oomycete infection. Ultracentrifugation-immunoblot experiments were used to show that RFP-tagged effectors were present in the endosome (GFP) fractions. Since tagged HARP1 caterpillars are not technically feasible, a HARP1 antibody could be developed for this experiment or HARP1 could be detected by mass spectrometry. Detection of native HARP1 in labeled endosomes would greatly alleviate the technical concerns regarding exogenous HARP1 application.

Response 1: Thank you for the comment. We agree that in this research work, the direct evidence of the native HARP1 entry into plant leaves via endocytosis is missing. This imperfection is largely due to the limited technique and is a common problem in the molecular research of insect pests in the present. The test which you suggested is hard to perform. Unlike the effector of pathogen can be amplified with the growth of pathogen in plant, insect effector (HARP1), which is secreted to plant tissues during herbivore, only exists around the wounding site and cannot be amplified. Therefore, if we should perform the similar test, much more materials are required and it is too difficult to obtain and I'm not sure whether the insect effector could be detected following by multiple extraction steps regardless of western blot or LC-MS.

Nevertheless, we are very appreciated for your advice. We designed an alternative experiment to analyze whether native HARP1 import requires endocytosis. In detail, we firstly treated the wounded leaf discs with A23 and then transferred the discs to incubate with V-HARP1 and OS. At last, the imported V-HARP1 and native HARP1 were analyzed by immunoblot, and the imported native

HARP1 was further confirmed by whole amount immunohistochemistry.

We put a lot of efforts into this experiment. Large amount of OS is required for this experiment and it takes four people about one week to collect the enough OS from cotton bollworm larvae. Furthermore, we spent quite a lot of time to optimize the experimental conditions. Finally, we finished the experiment and got the encouraging results. In the immunoblot assay, the imports of both V-HARP1 and the native HARP1 were reduced to an undetectable level by the pretreatment of A23 (Fig. 2d). The whole amount immunohistochemistry reveals that the signals of HARP1 from OS-incubated leaves were much weaker in the A23 pretreated samples compared to those A23 free samples (Fig. 2e). These results suggest that the import of native HARP1 depends on endocytosis, which is consistent with the import of recombinant V-HARP1.

Comment 2: Line 37-38: The text should be introduce elicitors in general many of which can elicit JA, not just FACs.

Response 2: Thank you for the comment. We have added some introduction of other insect elicitors in this new revision (Line 38-43, highlighted in blue).

Comment 3: Line 171: Should be "bimolecular" not "bilayer"

Response 3: We have made the correction. Thank you!

Comment 4: Line 317: Please add accession number XP_047035071 in addition to the URL

Response 4: We have added the accession number. Thank you!

Comment 5: Line 714 and many other places: "sights" should be "sites"

Response 5: We have made the correction. Thank you!

Reviewer #2

Comment1: However, with some of the responses I am not satisfied:

Response 3: The explanation for choosing the candidate genes is rather strange, authors just pick some proteins involved in membrane homeostasis and vesicle trafficking, without clear justification of any biological reasoning for that. It should be stated in the manuscript why these genes we chosen in particular and not the other (even the ones from the same family). Also authors claim the inability to produce and test the *ctl1/pat2/tet8* triple mutant due to the close proximity of the genes on the first chromosome. This problem can of course be avoided by using the CRISPR technology to mutate the genes (even all three at once...).

Response 1: Thank you for the comment. Yes, we agree that it might be more perfect to use *ctl1/pat2/tet8* triple mutant in this study. Our key conclusion is that an effector of caterpillar gets into plant cells via protein-mediated endocytosis. The coordination of these endocytosis related proteins is important to study the vesicle trafficking and the *ctl1/pat2/tet8* triple mutant might helpful for new findings. I'm also curious about this. However, we have provided sufficient data to support the main conclusion of this paper even without this triple mutant and generating the triple mutant via CRISPR technology requires a long period of time.

Comment 2: Response 4: The feeding experiment should be performed at once using all the possible genotypes and the same conditions (regarding the length of the feeding time). Limited growth space should not be a reason not to perform the experiment in the proper way.

Response 2: Thank you for the comment. Our intention is to compare the insect resistance between the wild type and the mutants. Therefore, we still think there is no problem to test the insect resistance of these mutants independently. Nevertheless, we try to provide a more acceptable response to this comment. We performed the feeding test using all the selected mutants and similar result was obtained (Supplementary Figure 10).

Reviewer #4

Comment 1: Both I and another reviewer are concerned that the authors used a concentration of HARP1 that is much higher than what normally would be found in caterpillar spit. This is required to see an effect in the authors' assays. Effects that are not physiologically relevant can occur when such a large excess of protein is used for experiments. The authors say that the high HARP1 concentration was needed because the protein was purified from bacteria and therefore may not have the right insect-specific modifications. I don't think that this is a sufficient response. If that is the hypothesis, then the authors should purify their protein from a lepidopteran cell expression system (e.g., Sf9 or High 5 cells are commonly used for this) and show that there is greater activity than when the protein is purified from bacteria. If the "most fitting buffer solution" of insect oral secretions is further required to obtain activity, then the transgenically produced fusion proteins can be added to insect oral secretions for the assays.

Response 1: Thank you for the comment. We agree that in this research work, the direct evidence of the native HARP1 entry into plant leaves via endocytosis is missing. This imperfection is largely due to the limited technique and is a common problem in the molecular research of insect pests in the present. We designed an experiment to detect whether native HARP1 import requires endocytosis. In detail, we firstly treated the wounded leaf discs with A23 and then transferred the discs to incubate with V-HARP1 and OS. At last, the imported V-HARP1 and native HARP1 were analyzed by immunoblot, and the imported native HARP1 was further confirmed by whole amount immunohistochemistry.

We put a lot of efforts into this experiment. Large amount of OS is required for this experiment and it takes four people about one week to collect the enough OS from cotton bollworm larvae. Furthermore, we spent quite a lot of time to optimize the experimental conditions. Finally, we finished the experiment and got the encouraging results. In the immunoblot assay, the imports of both V-HARP1 and the native HARP1 were reduced to an undetectable level by the pretreatment of A23 (Fig. 2d). The whole amount immunohistochemistry reveals that the signals of HARP1 from OS-incubated leaves were much weaker in the A23 pretreated samples compared to those A23 free samples (Fig. 2e). These results suggest that the import of native HARP1 depends on endocytosis, which is consistent with the import of recombinant V-HARP1.

We agree with you that we lack the evidence to support the idea about "most fitting buffer solution" though it sounds reasonable. So we deleted these discussions in the new revision. In addition, when leaf discs were incubated with 0.01mg/ml V-HARP1 for 2 hours, the V-HARP1 can be detected by immunoblot (Fig. 2d) but can be hardly detected by confocal microscopy (Supplementary Figure 1).

So the limited sensitivity of confocal detection requires a high concentration of V-HARP1. And in this revision, we give the evidence that import of native HARP1 depends on endocytosis, which is consistent with the observation of recombinant V-HARP1 import (Fig.2 d-e).

Comment 2: The author response “4, The manuscript was carefully examined and the grammar and spelling errors were collected.” does not inspire great confidence in their approach to correcting grammar and spelling errors.

Response 2: We have made some corrections in this new revision (highlighted in blue). Thank you!

REVIEWERS' COMMENTS

Reviewer #1 (Remarks to the Author):

The revision now includes new experiments where native HARP1 from applied *H. armigera* (Ha) oral secretions (OS) is detected by Western blot / in-situ hybridization, not just bacterially-expressed purified V-HARP1. Exogenously applied HARP1 protein (untagged component of total OS) is taken up in the margins of a wound, and A23 endocytosis inhibitor can inhibit the signal in both the Western and in-situ readouts. This gives further confidence in the conclusion of HARP1 cellular uptake at native concentrations. The inclusion of an OS treatment strengthens confidence that uptake would occur in a biological interaction.

The anti-HARP1 antibody was generated and validated in a previous publication, Chen et al 2019 PNAS. In that work the authors verified specificity through overexpression in plants, and showed that HARP1 protein levels in OS can be silenced by dsRNA method.

The rebuttal letter is very persuasive in explaining why experiments with the herbivore itself would be difficult, given tiny amounts of HARP1 present in a native interaction. I would argue that this is further reason for caution in interpreting previous results from artificially high concentrations, and thus it is great to see the new OS results!

Overall, I find the twice revised work to be a large step forward for cell biology of chewing herbivore effectors. The identification of critical protein regions for uptake may have major impacts beyond *Ha-Arabidopsis* interactions, despite technical concerns driven by limitations of the system.

Reviewer #4 (Remarks to the Author):

The authors partially addressed my previous comment (and that of Reviewer 1) by showing that native HARP1 from caterpillar saliva enters plant cells by endocytosis. This does not fully address the question of whether the phenotypic effects that are induced by synthesized V-HARP1 are also induced by the lower concentrations of HARP1 that are found in caterpillar saliva. Nevertheless, the endocytosis of native HARP1 is an important addition to this manuscript.

I was not able to find the collection of caterpillar saliva for this experiment described in the methods section. Nor was there a description of how the caterpillar secretions were added to leaf disks. There is a description of treatment with V-HARP1 solution added in blue, but this does not describe treatment with native HARP1 from collected caterpillar saliva.

Response to reviewers

Thank you for your suggestions during the three rounds of review. With your help we designed and performed new experiments and the results largely strengthen the confidence of the main conclusion of this study. Thank you!

Response to Reviewer #1:

The comments you provided are greatly appreciated. Your valuable insights have offered us numerous constructive ideas to enhance our manuscript. Thank you!

Response to Reviewer #4:

Comment: I was not able to find the collection of caterpillar saliva for this experiment described in the methods section. Nor was there a description of how the caterpillar secretions were added to leaf disks. There is a description of treatment with V-HARP1 solution added in blue, but this does not describe treatment with native HARP1 from collected caterpillar saliva.

Response: Thank you for the comments! In this new revision, the OS collection and the detailed method of whole amount immunohistochemistry were added (page 16). Thank you again for your helpful advises during the three rounds of review.